# Insights into the key determinants of membrane protein topology enable the identification of new monotopic folds

**Sonya Entova[1], Jean-Marc Billod[2], Jean-Marie Swiecicki[1], Sonsoles Martín-Santamaría[2], Barbara Imperiali[1,3]***

[1]Department of Biology, Massachusetts Institute of Technology, Cambridge, United States; [2]Department of Structural & Chemical Biology, Centro de Investigaciones Biológicas, Madrid, Spain; [3]Department of Chemistry, Massachusetts Institute of Technology, Cambridge, United States

**Abstract** Monotopic membrane proteins integrate into the lipid bilayer via reentrant hydrophobic domains that enter and exit on a single face of the membrane. Whereas many membrane-spanning proteins have been structurally characterized and transmembrane topologies can be predicted computationally, relatively little is known about the determinants of membrane topology in monotopic proteins. Recently, we reported the X-ray structure determination of PglC, a full-length monotopic membrane protein with phosphoglycosyl transferase (PGT) activity. The definition of this unique structure has prompted in vivo, biochemical, and computational analyses to understand and define key motifs that contribute to the membrane topology and to provide insight into the dynamics of the enzyme in a lipid bilayer environment. Using the new information gained from studies on the PGT superfamily we demonstrate that two motifs exemplify principles of topology determination that can be applied to the identification of reentrant domains among diverse monotopic proteins of interest.

DOI: https://doi.org/10.7554/eLife.40889.001

*For correspondence:
imper@mit.edu

**Competing interests:** The authors declare that no competing interests exist.

## Introduction

Membrane proteins represent an essential and diverse component of the proteome. Our understanding of how integral membrane proteins are folded and inserted into the membrane continues to evolve with the development of more sophisticated structural, biochemical, and computational tools. The topology of integral membrane proteins is defined by how many times, and in which direction, the sequence spans the lipid bilayer: polytopic membrane proteins span the membrane multiple times, bitopic membrane proteins span the membrane a single time, and monotopic membrane proteins do not span the membrane, but instead are embedded in the membrane via a reentrant domain that enters and exits on a single face of the membrane (*Blobel, 1980*). Current bioinformatics approaches (*Elazar et al., 2016*; *Krogh et al., 2001*; *Tsirigos et al., 2015*) enable a relatively reliable prediction of transmembrane helix topology in polytopic and bitopic membrane proteins on the basis of hydrophobicity, homology with known protein structures, and the 'positive-inside rule', by which membrane proteins have been empirically determined to preferentially adopt topologies that position positively charged residues at the cytoplasmic face of the membrane (*Gafvelin and von Heijne, 1994*; *Heijne, 1986*; *von Heijne, 2006*), among other parameters. By comparison, current knowledge of monotopic topologies is relatively limited. In particular, the key determinants that distinguish reentrant domains in monotopic proteins from transmembrane helices in bitopic proteins, to result in the two distinct topologies, are poorly understood. Here we present an in-depth analysis of two conserved motifs that are key determinants of a monotopic topology in a

membrane-bound phosphoglycosyl transferase (PGT) enzyme with a single reentrant domain. We anticipate that these two motifs embody common themes in reentrant domain formation across diverse monotopic membrane proteins, and herein demonstrate how they can be used to distinguish such domains from transmembrane helices at a protein sequence level.

PGTs are integral membrane proteins that initiate a wide variety of essential glycoconjugate biosynthesis pathways, including peptidoglycan and N-linked glycan biosynthesis, by catalyzing the transfer of a phosphosugar from a sugar nucleoside diphosphate donor to a membrane-resident polyprenol phosphate. PGTs can be grouped broadly into two superfamilies based on their membrane topology (*Lukose et al., 2017*). One superfamily, exemplified by bacterial MraY and WecA, is composed of polytopic PGTs with 10- and 11- transmembrane helices respectively and active sites crafted from extended cytoplasmic inter-TM loops (*Anderson et al., 2000*). The eukaryotic PGT Alg7, which initiates the dolichol pathway for N-linked protein glycosylation, belongs to this superfamily (*Burda and Aebi, 1999*). A second superfamily is exemplified by the monotopic enzyme, PglC, from the Gram-negative bacterium *Campylobacter jejuni* (*Glover et al., 2006*; *Szymanski et al., 1999*). The PGTs in this superfamily share a common functional core, which is homologous to PglC and comprises a single N-terminal membrane-inserted domain and a small globular domain (*Lukose et al., 2015*). The superfamily also includes WbaP, which features a PglC-like core elaborated with four N-terminal transmembrane helices that are not necessary for catalytic activity (*Saldías et al., 2008*), and the bifunctional enzyme PglB from *Neisseria gonorrhoeae*, which has an additional C-terminal aminotransferase domain (*Hartley et al., 2011*). Topology predictions using multiple algorithms suggested that the N-terminal hydrophobic domain of PglC, and the analogous domain in other superfamily members, forms a transmembrane helix, such that the N-terminus is in the periplasm and the globular domain in the cytoplasm (*Furlong et al., 2015*; *Lukose et al., 2017*). However, biochemical analysis of WcaJ, a WbaP homolog, supported a model wherein both termini of the corresponding domain of WcaJ are in the cytoplasm, forming a reentrant topology, rather than a membrane-spanning one (*Furlong et al., 2015*).

Recently, we reported the X-ray structure of PglC from *Campylobacter concisus* to a resolution of 2.7 Å (PDB 5W7L) (*Ray et al., 2018*). The structural analysis complements previous biochemical studies on PglC from *C. jejuni*; the homologs share 72% sequence identity. In the reported structure, the N-terminal domain of PglC forms a reentrant helix-break-helix structure, termed the reentrant membrane helix (RMH), such that both the N-terminus and the globular domain, which includes the active site, are on the cytoplasmic face of the inner membrane (*Figure 1A*). A reentrant topology was further confirmed in vivo using a substituted cysteine accessibility method (SCAM) (*Nasie et al., 2013*; *Ray et al., 2018*). As such, the same reentrant topology is confirmed in both PglC and the corresponding domain of an elaborated WbaP homolog, suggesting that the topology and mechanism of membrane association enforced by the RMH is a conserved feature of the PGTs in this extensive superfamily.

The RMH, as the only membrane-resident domain of PglC, plays a crucial role in anchoring PglC in the membrane. The RMH also interacts with a coplanar triad of amphipathic helices to position the active site of PglC at the membrane-water interface, thereby enabling efficient phosphosugar transfer from a soluble nucleotide diphosphate donor to a membrane-resident polyprenol phosphate acceptor. The RMH is composed of an α-helix broken at a 118° angle by a conserved Ser23-Pro24 dyad (*Figure 1A–C*). In the reported structure, Pro24 disrupts the hydrogen-bonding network of the RMH backbone, creating a break in the helix. This break is stabilized in turn by the orientation of the Ser23 side chain, which forms a 2.6 Å hydrogen bond with the backbone carbonyl of Ile20, thus satisfying one of the backbone hydrogen bonds lost due to Pro24. Pro24 is highly conserved among PglC homologs, and has been previously shown to be important for PglC activity (*Lukose et al., 2015*). At the N-terminus of PglC there is a similarly conserved Lys7-Arg8 dyad. Lys7 makes short-range contacts with the C-terminus of the globular domain via residue Asp169, and Arg8 interacts with the headgroup of a co-purified phospholipid molecule (*Figure 1D*).

Most structurally-characterized examples of ordered reentrant helices occur in polytopic membrane proteins in which topology is largely defined by the presence of multiple transmembrane helices (*Viklund et al., 2006*) In contrast, the RMH of monotopic PglC is the sole determinant of the reentrant topology. Thus, PglC provides a unique opportunity to identify structural motifs that influence helix geometry to result in membrane-inserted domains that favor a reentrant topology over a transmembrane one. Many of the structurally-characterized monotopic membrane proteins associate

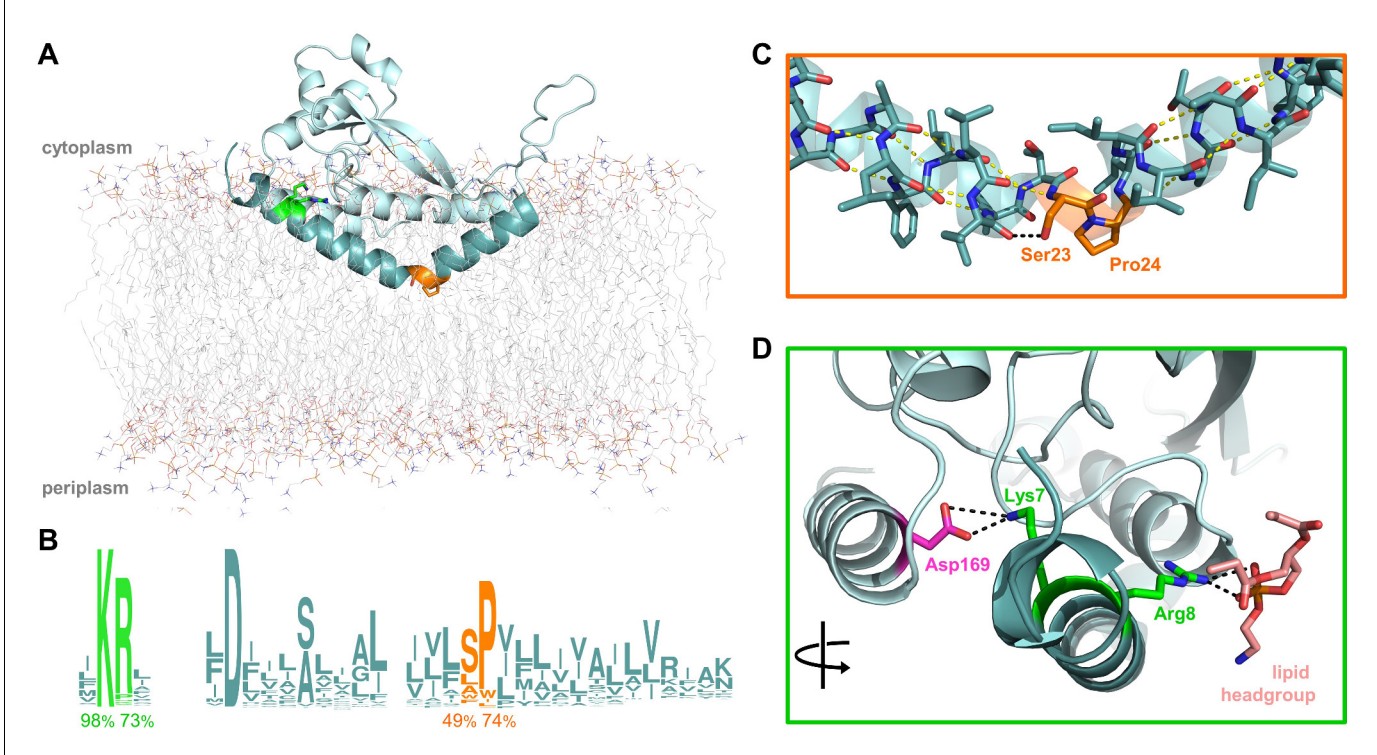

**Figure 1.** Overview of PglC highlighting the two conserved motifs in the RMH. (**A**) Model of PglC from *C. concisus* in a POPE lipid bilayer. The RMH is shown in teal, Lys7 and Arg8 as green sticks, Ser23 and Pro24 as orange sticks. (**B**) Sequence logo showing conservation in the RMH domain among PglC homologs. Percent conservation is noted below each residue of interest. Logo generated using weblogo.berkeley.edu. (**C**) Detailed view of the Ser-Pro motif. Pro24 disrupts the hydrogen-bonding network (yellow dashes) of the RMH backbone. A 2.6 Åhydrogen bond (black dashes) is formed between the hydroxyl side group of Ser23 and the backbone carbonyl of Ile20. (**D**) Detailed view of the Lys-Arg motif. Lys7 forms a salt bridge with Asp169 (magenta). Arg8 interacts with a the headgroup of a copurified lipid (salmon).

DOI: https://doi.org/10.7554/eLife.40889.002

The following figure supplement is available for figure 1:

**Figure supplement 1.** PglC is stable in a model POPE lipid bilayer.
DOI: https://doi.org/10.7554/eLife.40889.003

with the membrane through hydrophobic loops or amphipathic helices (*Bracey et al., 2004*). However, PglC is currently the only example of a structurally-characterized monotopic membrane protein that associates with the membrane via a highly-ordered, reentrant helix-break-helix motif that penetrates into the hydrophobic membrane core.

Current membrane topology prediction algorithms typically default to the assumption that hydrophobic α-helices of a certain length range are membrane-spanning (*Elazar et al., 2016*; *Krogh et al., 2001*; *Tsirigos et al., 2015*), making the specific prediction of monotopic topologies with reentrant α-helices challenging. Thus, delineating drivers of reentrant topology would enable accurate predictions of this topology among membrane proteins, and would provide insight into other membrane proteins similarly identified as having a single membrane-inserted domain, including the eukaryotic scaffolding protein caveolin-1, known to be monotopic (*Aoki et al., 2010*; *Okamoto et al., 1998*), and the mammalian membrane-bound enzyme diacylglycerol kinase ε (*Epand et al., 2016*). Diacylglycerol kinase ε, like PglC, has a single N-terminal hydrophobic domain, but its membrane topology remains unclear (*Decaffmeyer et al., 2008*; *Nørholm et al., 2011*). In the current study, we apply in vivo, biochemical, and computational methodologies to identify two conserved motifs that both drive formation of the RMH in PglC and contribute to the stability of the final fold, enforcing a monotopic topology for PglC. The significance of proper RMH formation for catalytic activity is also discussed. Importantly, we demonstrate that the principles of RMH formation identified in PglC are broadly generalizable by using them to identify a reentrant topology in LpxM,

a fatty acid acyltransferase involved in the biosynthesis of lipid A. LpxM represents a large family of enzymes, unrelated to PglC, which was previously predicted to be bitopic.

## Results

### The reported structure of PglC is stable in a membrane environment

The crystal structure of PglC shows that the N-terminal hydrophobic domain forms a reentrant membrane helix (RMH) that anchors the fold in the membrane (*Ray et al., 2018*). However, as the reported structure was generated using detergent-solubilized PglC in an aqueous environment, we applied molecular dynamics (MD) simulations to investigate whether the structure would be stable in a more native membrane-like environment. A model of the reported structure of PglC from *C. concisus* in a 1-palmitoyl-2-oleoyl-*sn*-glycero-3-phosphoethanolamine (POPE) membrane was generated computationally (*Lomize et al., 2012*) (*Figure 1A*). The resulting model supports the interaction of PglC with the membrane via the RMH, which penetrates the membrane through one of the lipid leaflets. This model was further submitted to MD simulations to assess the stability of PglC in a membrane environment. Over 400 ns of simulation time, RMSD values within ~1 Å were measured for PglC (*Figure 1—figure supplement 1*). These analyses support that the conformation of PglC observed in the crystal structure reflects a native state in a lipid bilayer environment.

### A conserved Ser-Pro motif is a key determining factor of reentrant topology in PglC

The conservation of the Ser-Pro motif and its location at the break in the RMH suggested that the motif plays an important role in establishing a reentrant topology in PglC. Thus, the significance of both residues in topology determination was evaluated in vivo by the same SCAM analysis (*Nasie et al., 2013*) used previously to assess reentrant topology in PglC (*Ray et al., 2018*) and in the elaborated PglC superfamily member, WcaJ (*Furlong et al., 2015*). For topology analysis by SCAM, the subcellular localization of unique cysteines introduced into a target protein is determined by whether such cysteines react in vivo with the cell-permeable reagent N-ethylmaleimide (NEM) or with 2-sulfonatoethyl methanethiosulfonate (MTSES), permeable only to the outer membrane (*Figure 2A*). Reaction with either reagent 'protects' the cysteine from subsequent labeling by methoxypolyethylene glycol maleimide (PEG-mal, MW 5 kDa) following cell lysis. Cysteines located in the periplasm are thus distinguished from cytoplasmic cysteines by protection of the former from PEGylation by both NEM and MTSES, whereas cytoplasmic cysteines are PEGylated following treatment with MTSES but not following treatment with NEM. PEGylation is determined by a 5–10 kDa shift to a higher molecular weight band in Western blot analysis relative to the native protein. PglC from *C. jejuni* was used for all SCAM analysis and in vitro assays described herein. Corresponding key residues in PglC from *C. jejuni* and *C. concisus* are listed in *Supplementary file 1*.

In a previously reported analysis of wild-type PglC topology, two cysteine substitutions at the N-terminus of PglC (K4C and F6C) and two in the globular domain (S88C and S186C) were all found to be cytoplasmic, indicating a reentrant topology (*Figure 2B*). Importantly, all four cysteine-substituted PglC variants were found to retain 10–50% catalytic activity relative to wild-type PglC (*Figure 2—figure supplement 1*). In the current study, the same cysteine substitutions were used in SCAM analyses to determine the topology of S23A and P24A PglC variants. Similar to wild-type PglC, both variants were found to form reentrant topologies with both termini in the cytoplasm (*Figure 2C*). However, in a S23A/P24A PglC variant, all four cysteines were significantly protected from PEGylation by both NEM and MTSES, suggesting that all four cysteines were in the periplasm. Although the precise nature of the S23A/P24A PglC topology cannot be determined from these data, it is clear that the folding of this variant is substantially perturbed by mutation of the Ser-Pro dyad to Ala-Ala. These results suggest that Ser23 and Pro24 act synergistically to establish a reentrant topology that positions both the N- and C-terminus of PglC in the cytoplasm.

Whereas the crystal structure of PglC provides a 'snapshot' of molecular interactions at the helix break in the RMH, MD simulations supplied a more dynamic view. Indeed, in simulations of PglC performed in a POPE membrane, it was observed that the break imposed by the Ser-Pro motif is additionally maintained by hydrogen bonding alternately between the side chain hydroxyl group and backbone amide of Ser23 and the backbone carbonyls of Leu19 and Ile20 (*Figure 3A*). This suggests

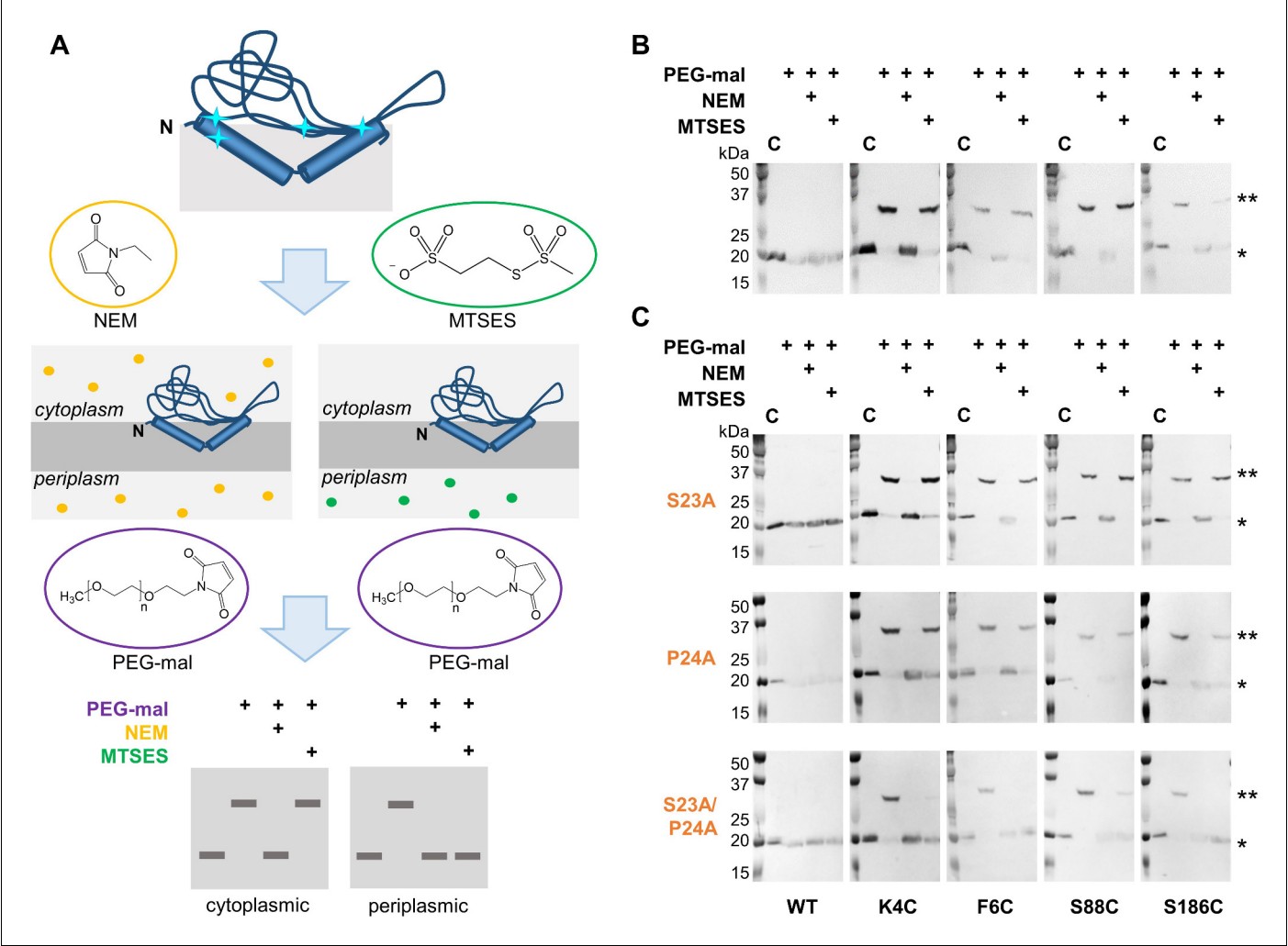

**Figure 2.** Ser23 and Pro24 act together to enforce the reentrant PglC topology. (**A**) Schematic representation of SCAM analysis used to assess the topology of wild-type PglC and variants. Cyan starbursts (top) indicate the location of unique cysteines introduced into PglC. (**B**) SCAM analysis of wild-type PglC topology (* = native PglC; ** = PglC labeled with PEG-mal; C = control, no PEG-mal labeling). (**C**) SCAM analysis of S23A, P24A and S23A/P24A PglC variant topologies. All SCAM experiments were performed in duplicate or more. Representative Western blots are shown.

DOI: https://doi.org/10.7554/eLife.40889.004

The following figure supplement is available for figure 2:

**Figure supplement 1.** PglC Cys variants used for SCAM analyses.

DOI: https://doi.org/10.7554/eLife.40889.005

that the break in the RMH is stabilized by an extensive network of dynamic hydrogen bonds between Leu19, Ile20 and Ser23 that compensate for the absence of a hydrogen-bond donor in Pro24 and enforce a helix-break-helix topology.

The high overall hydrophobicity of the RMH domain near the N-terminus of PglC likely results in targeting of the nascent polypeptide to the membrane early in translation (*Martoglio and Dobberstein, 1998*). Therefore, we hypothesized that the break in the RMH, enforced by the Ser-Pro motif and the resulting hydrogen-bonding network described above, could allow the N-terminal RMH to form independently of the C-terminal globular domain and insert the membrane in a reentrant topology. To investigate further the role that this motif might play in such early folding events, peptides representing the RMH of both wild-type and S23A/P24A PglC were examined in MD folding simulations (*Figure 3B*). Both peptides were initially generated in an extended conformation and allowed to fold for 100 ns in water to simulate early co-translational folding events. Next, a shift to a more hydrophobic medium (20% isopropanol in water) for a further 1.5 μs provided an environment

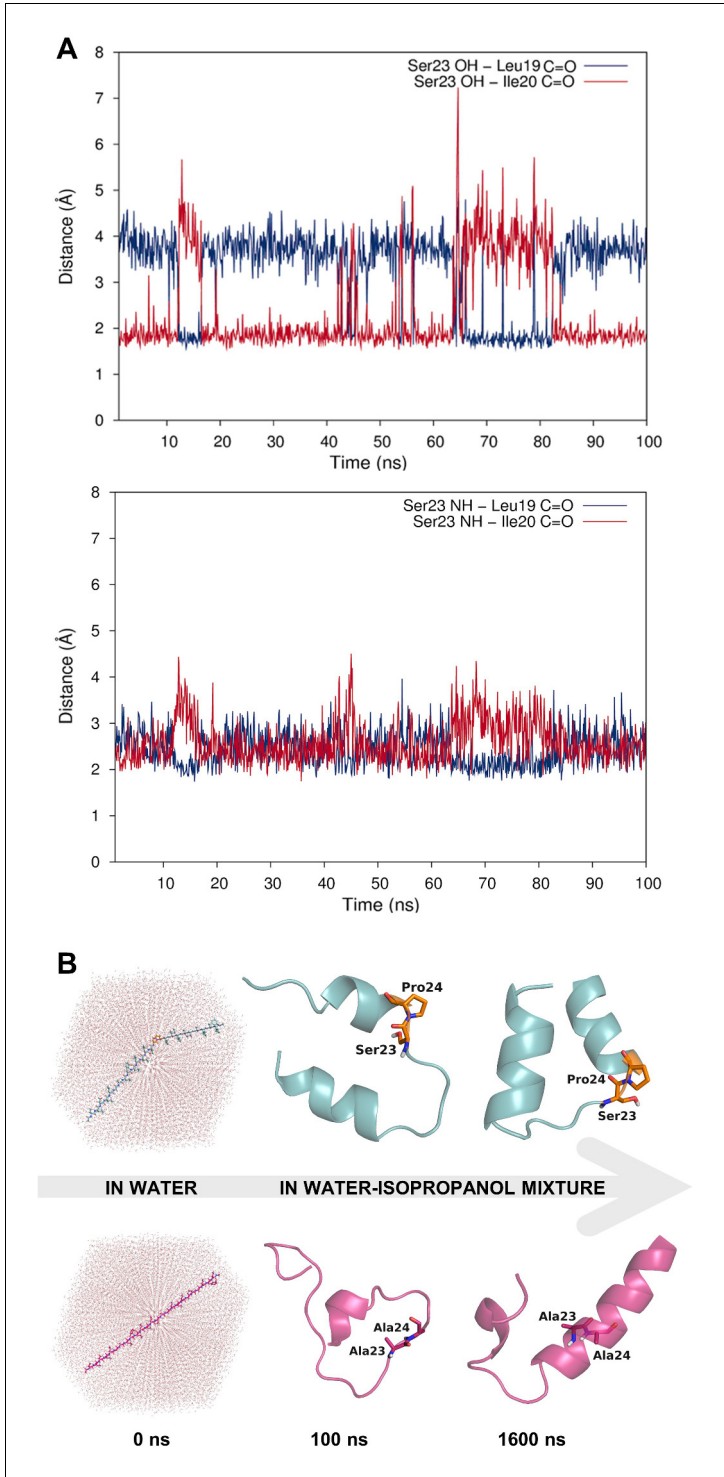

**Figure 3.** Dynamic hydrogen bonding around the Ser-Pro motif facilitates early folding of the RMH domain. (A) Dynamic hydrogen bonding between the side chain hydroxyl (top panel) or backbone amide-NH (bottom panel) of Ser23 and the backbone amide-C=O of Leu19 and Ile20 measured during MD simulations of PglC in a POPE lipid bilayer. (B) Peptides corresponding to the RMH domains of wild-type (top) and S23A/P24A PglC (bottom) were folded from an extended conformation (left panel) for 100 ns in water (middle panel), followed by an additional 1500 ns in 20% isopropanol/water (right panel).
DOI: https://doi.org/10.7554/eLife.40889.006

The following figure supplement is available for figure 3:

*Figure 3 continued on next page*

*Figure 3 continued*
**Figure supplement 1.** The Ser-Pro motif facilitates formation of the RMH domain.
DOI: https://doi.org/10.7554/eLife.40889.007

more closely resembling the membrane. Significant differences in folding behavior between the two peptides were observed over the full 1.6 μs of simulation time; in the peptide corresponding to wild-type PglC, residues Leu19 through Pro24, encompassing the Ser-Pro motif and surrounding hydrogen bonds, remained mostly unstructured, while the N-terminus and the sequence following the Ser-Pro motif rapidly adopted an α-helical conformation (*Figure 3B* and *Figure 3—figure supplement 1*). In contrast, in the peptide corresponding to S23A/P24A PglC, residues Leu19 through Ile33 folded into a continuous α-helix, and this secondary structure appeared much later in the folding process. These simulations indicate that the Ser-Pro motif drives formation of the helix-break-helix structure observed in the RMH of wild-type PglC, while the corresponding domain of S23A/P24A PglC has an intrinsic tendency to form a single, uninterrupted helix. The difference in the simulated folding of the two peptides underscores the significance of the Ser-Pro motif for RMH formation, and further supports a model by which this motif facilitates proper folding and membrane insertion of the N-terminal domain in a reentrant topology during an early co-translational folding event.

## The Ser-Pro motif contributes to the overall stability of the PglC fold

The SCAM analyses and MD simulations suggest a role for the Ser-Pro motif in early formation of the N-terminal RMH domain. However, additional analyses were needed to elucidate the contribution of the motif to stabilizing the overall PglC fold. To illustrate the contribution of the Ser-Pro dyad to the overall stability of PglC, in vitro thermal stability measurements were performed on purified wild-type and S23A/P24A SUMO-PglC variants (*Figure 4*). Incorporation of an N-terminal SUMO tag has been previously reported to aid greatly in purification of PglC (*Das et al., 2016*; *Lukose et al., 2015*; *Ray et al., 2018*) and SUMO-PglC has been confirmed to be catalytically active (*Das et al., 2016*; *Lukose et al., 2015*) and to adopt a native reentrant topology (*Figure 4—figure supplement 1*). Thermal stability could not be measured by circular dichroism (CD) as it was observed that S23A/P24A SUMO-PglC does not purify to homogeneity (*Figure 4—figure supplement 2*), therefore, CD analysis on such a sample would be confounded by signal from co-purifying contaminants. It has also been noted that unfolding of α-helical membrane proteins is often driven through loss of tertiary contacts rather than secondary structure (*Grinberg et al., 2001*; *Stowell and Rees, 1995*; *Vogel and Siebert, 2002*), making CD spectroscopy less informative for thermal stability analysis. Thus, a thermal shift assay that specifically reports on stability as a function of resistance to precipitation upon heating, recently applied to the polytopic membrane protein dolichylphosphate mannose synthase (*Gandini et al., 2017*), was used. Following heating, precipitated protein was removed by centrifugation and the soluble fraction quantified by gel densitometry to determine a $T_m$ for both variants. It was found that the S23A/P24A SUMO-PglC variant, with a $T_m$ of 42.6 ± 2.0 °C, is significantly less stable relative to wild-type SUMO-PglC, which had a $T_m$ of 49.9 ± 1.5 °C (*Figure 4A*). The $\Delta T_m$ of 7.3 ± 2.5 °C between the wild-type and S23A/P24A SUMO-PglC indicates a loss in thermal stability upon mutation of the Ser-Pro motif to Ala-Ala. Notably, the Ala-Ala substitution control variants I26A/L27A SUMO-PglC and K187A/E188A SUMO-PglC experienced only slight decreases in stability relative to wild-type, with $T_m$ values of 47.3 ± 0.9 and 46.6 ± 1.1 °C, respectively (*Figure 4—figure supplement 3*). Thus, the Ser-Pro motif is not only necessary for formation of the RMH domain, but additionally has a strong influence on the stability of the entire PglC structure.

To elucidate further the role that the Ser-Pro motif plays in maintaining the stability of the native PglC fold, MD simulations were performed on both wild-type PglC and a S23A/P24A variant, generated in silico by substituting the Ser-Pro motif with Ala-Ala. Whereas wild-type PglC was stable over 400 ns of simulations in a POPE membrane, the interior of S23A/P24A PglC, containing the putative polyprenol phosphate binding site proximal to the active site, appeared to collapse. Specifically, in S23A/P24A PglC, the positions of residues Leu21 on the RMH, and of Leu90 and Val180 on two amphipathic helices at the membrane interface, were all found to move closer to each other over the course of 400 ns, significantly reducing the volume of the protein interior (*Figure 4B* and

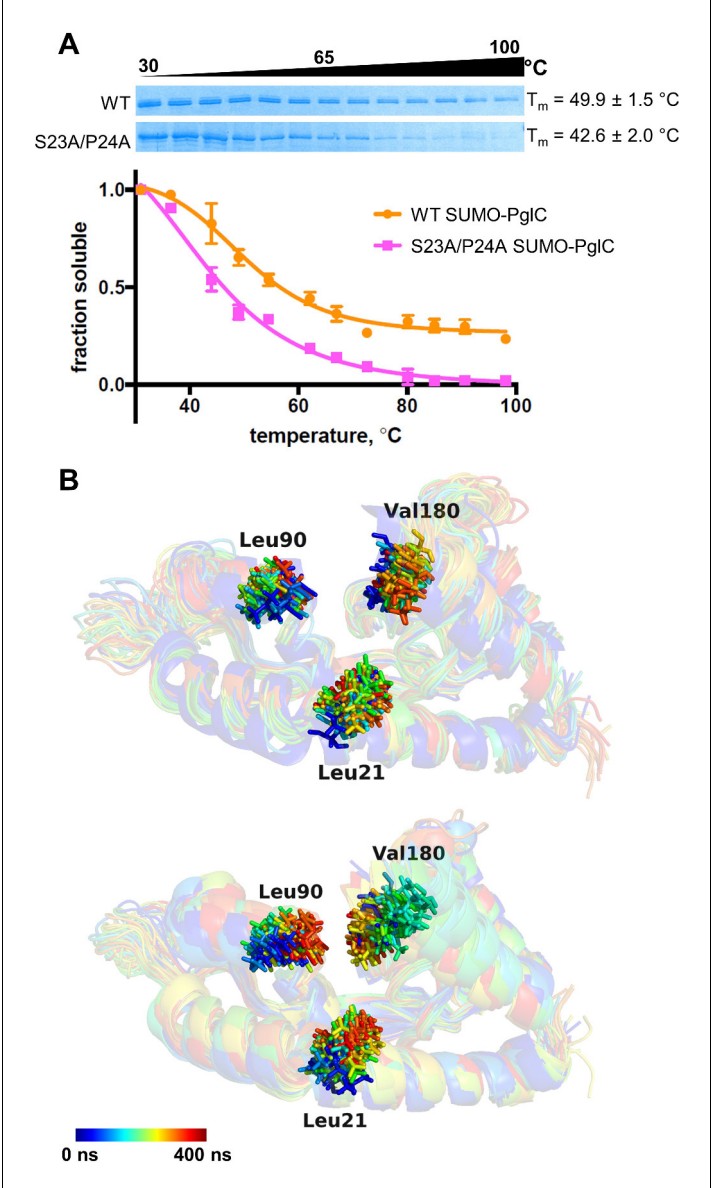

**Figure 4.** The Ser-Pro motif contributes to stability of the PglC fold. (**A**) Thermal shift analysis of wild-type and S23A/P24A SUMO-PglC. Error bars are given for mean ± SEM, n = 3. (**B**) Superimposition of frames, taken at 10 ns intervals, along MD simulations of wild-type (top panel) and S23A/P24A (bottom panel) PglC. Colored from blue, t = 0 ns to red, t = 400 ns. PglC is represented as a semi-transparent cartoon and residues Leu21, Leu90 and Val180 as sticks.

DOI: https://doi.org/10.7554/eLife.40889.008

The following figure supplements are available for figure 4:

**Figure supplement 1.** SUMO-PglC shows a reentrant topology similar to native PglC.
DOI: https://doi.org/10.7554/eLife.40889.009
**Figure supplement 2.** SDS-PAGE analysis of wild-type and S23A/P24A SUMO-PglC.
DOI: https://doi.org/10.7554/eLife.40889.010
**Figure supplement 3.** Thermal shift assays of control SUMO-PglC variants.
DOI: https://doi.org/10.7554/eLife.40889.011
**Figure supplement 4.** Mutation of the Ser-Pro motif causes a 'collapse' of the PglC fold interior.
DOI: https://doi.org/10.7554/eLife.40889.012
**Figure supplement 5.** Mutation of the Ser-Pro motif reduces lipid occupancy in the PglC fold interior.
DOI: https://doi.org/10.7554/eLife.40889.013

*Figure 4—figure supplement 4*). The positions of these same residues remained relatively unchanged in wild-type PglC. The collapse of the S23A/P24A PglC tertiary structure upon substitution of the Ser-Pro motif suggests that the motif plays an important role not only in forming the RMH, but also in maintaining the RMH and the associated amphipathic helices in the correct conformation in the PglC fold.

Additionally, it was observed that the collapse of the S23A/P24A PglC fold appeared to hinder the passage of lipids into the interior of the fold relative to wild-type PglC. At the start of MD simulations of both wild-type and S23A/P24A PglC, two phospholipid molecules were observed to occupy the fold interior. Both phospholipids continued to occupy this inner cavity during most of the simulation of wild-type PglC, whereas over the course of the simulation of S23A/P24A PglC, one phospholipid appeared to leave the cavity (*Figure 4—figure supplement 5*). As the fold interior contains the putative polyprenol phosphate binding site, lipid access into the PglC-fold interior is crucial for substrate binding. Thus, the role played by the Ser-Pro motif in maintaining the PglC fold also has immediate significance for catalytic activity.

## A conserved positively charged motif is also a determinant of reentrant topology

In addition to the Ser-Pro motif, two residues at the N-terminus of PglC, Lys7 and Arg8, are also highly conserved and participate in electrostatic interactions with the globular domain and surrounding phospholipids (*Ray et al., 2018*). We hypothesized that this Lys-Arg motif might also contribute to the reentrant topology of PglC. Thus, the topology of K7A and R8A PglC variants was evaluated

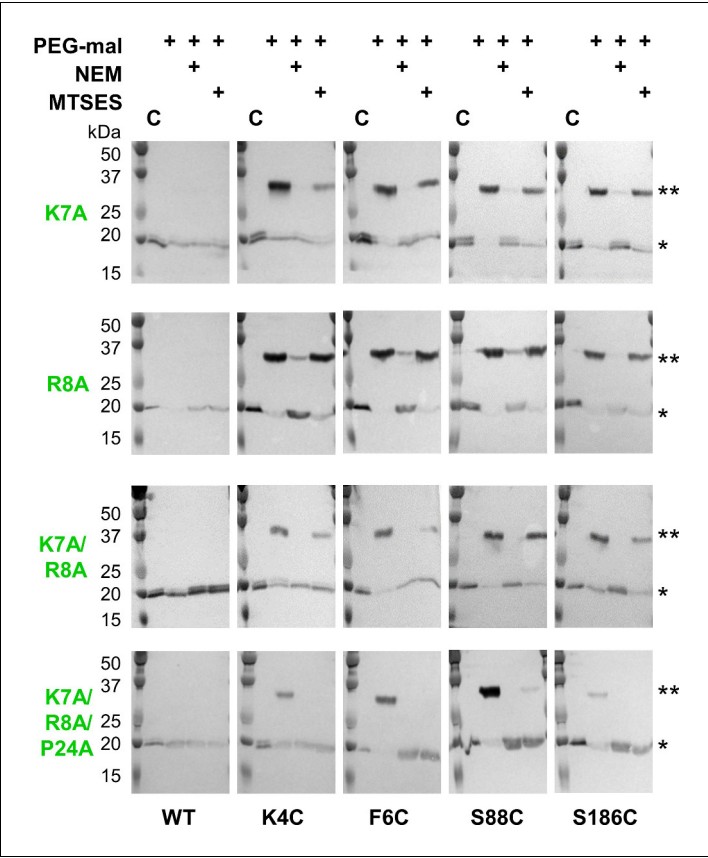

**Figure 5.** Lys7 and Arg8 additionally contribute to reentrant topology determination. SCAM analysis of K7A, R8A, K7A/R8A and K7A/R8A/P24A PglC variant topologies (* = native PglC; ** = PglC labeled with PEG-mal; C = control, no PEG-mal labeling). All SCAM experiments were performed in duplicate or more. Representative Western blots are shown.

DOI: https://doi.org/10.7554/eLife.40889.014

by SCAM (*Figure 5*). Mutation of either Lys7 or Arg8 to Ala did not have a significant effect on the final topology of PglC. However, when both conserved residues were mutated in the K7A/R8A PglC variant, it was observed that the N-terminus (represented by K4C and F6C) became partially localized to the periplasm. This suggested that the K7A/R8A PglC population was split between proteins adopting the native reentrant topology, in which the N-terminus is in the cytoplasm, and those adopting a non-native membrane-spanning topology. Notably, a K7A/R8A/P24A PglC variant adopted the same non-native, all-periplasmic topology previously observed for S23A/P24A PglC, suggesting that, as with S23A/P24A PglC, the native folding process of the RMH had been significantly perturbed. Taken together, these analyses indicate that the positively charged Lys-Arg motif and the helix-breaking Ser-Pro motif act cooperatively to enforce a reentrant topology for PglC.

## RMH geometry is held stable during catalysis

While the in vivo SCAM analyses and MD simulations are consistent with the reported structure of PglC, we noted that the inter-helical angle of the RMH in the structure (118°) (*Figure 1A*) is significantly more open than that observed in the peptide modeling (*Figure 3B*) and in reentrant helices found in reported polytopic membrane protein structures (*Viklund et al., 2006*). Therefore, we performed cysteine crosslinking studies to determine whether the conformation of the RMH in the reported structure reflects the native PglC fold and to further probe the significance of this conformation in catalysis.

Two cysteines were introduced into PglC, one at the N-terminus and one in the globular domain, at positions Glu3 and Ile163 respectively. If the reported structure reflects a native conformation of the RMH, the two residues at these positions have a $C_\beta$-$C_\beta$ distance of ~5.2 Å in the PglC fold (*Figure 6A*). Although oxidation to a disulfide was not observed, the two cysteines could be crosslinked in the E3C/I163C SUMO-PglC variant with dibromobimane (bBBr), a short, bifunctional thiol-specific labeling reagent. An increase in fluorescence is reported in the literature for bBBr upon thiol-thiol crosslinking (*Kim and Raines, 1995*; *Kosower and Kosower, 1987*). Indeed, bBBr-treated E3C/I163C SUMO-PglC was significantly more fluorescent than were either single cysteine variants (E3C SUMO-PglC and I163C SUMO-PglC), or a control variant (E3C/S88C SUMO-PglC) with two cysteine residues too far apart for crosslinking by bBBr (*Figure 6B*). Crosslinking was quantified by fluorescence densitometry of bands corresponding to monomeric PglC on SDS-PAGE gels, ensuring that only intramolecular crosslinking was included in the analysis. These crosslinking studies suggest that the extended RMH observed in the crystal structure reflects a native conformation of the PglC fold.

To confirm that crosslinking was capturing a stable conformation of the RMH in E3C/I163C SUMO-PglC and not a transient intermediate, the position of the N-terminus relative to the globular domain was further investigated by MD. In simulations of PglC in a POPE membrane it was observed that the distance between these two residue positions remains invariable over 400 ns (*Figure 6C*). This supports a model of PglC in which the RMH is constrained in the position observed in the crystal structure, with little conformational freedom of the N-terminus relative to the globular domain. Importantly, it was found that intramolecular crosslinking of E3C/I163C SUMO-PglC by bBBr did not impact catalytic activity (*Figure 6D*). This indicates that the crosslinked conformation of the RMH domain, which places the N-terminus in close proximity to the globular domain, is compatible with catalysis.

## Applying insight from PglC topogenesis studies to other bacterial membrane proteins

The Membranome database of single-helix transmembrane proteins (http://membranome.org/) (*Lomize et al., 2017*) contains curated data on >6000 known and predicted bitopic membrane proteins from eukaryotic and prokaryotic genomes. We manually parsed the Membranome list of bacterial bitopic proteins, comprising 196 sequences from *E. coli*, for hydrophobic domains that might be reentrant based on the following criteria: (1) the sequence contains an N-terminal hydrophobic domain that contains conserved helix-breaking residues and is preceded by conserved positive charges; (2) for candidates of known function, a reentrant topology would be consistent with the reported biological role; (3) where possible, covariance analysis confirms probable contacts between the N-terminal hydrophobic domain and the rest of the fold. Covariance analysis on sequence

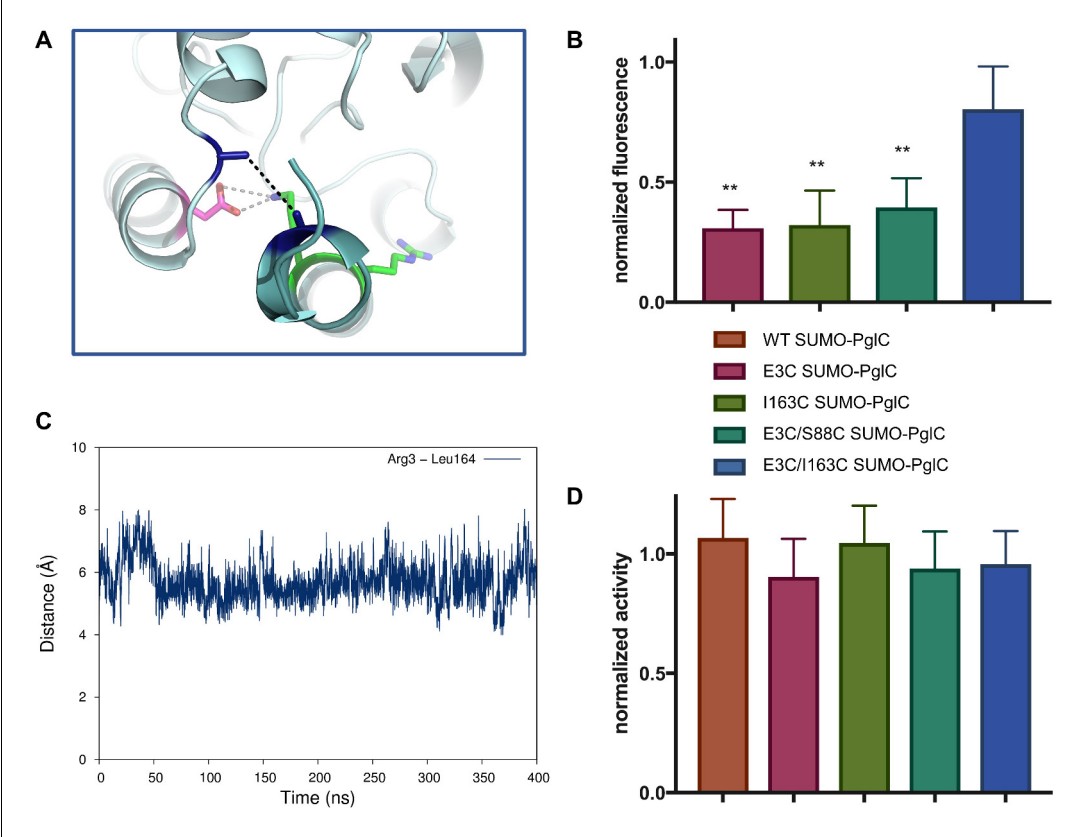

**Figure 6.** The RMH is held in the observed conformation during catalysis. (**A**) Detailed view showing the location of bBBr crosslinking (black dashes). The $C_\alpha$ and $C_\beta$ of Arg3 and Leu164 in the structure of PglC from *C. concisus* are shown as dark blue sticks (the remainder of the side chains is omitted for clarity). The corresponding residues Glu3 and Ile163 in PglC from *C. jejuni* were substituted with Cys for bBBr crosslinking studies. (**B**) Fluorescence of SUMO-PglC variants following crosslinking with bBBr, normalized to fluorescence of DTT-quenched samples (quenched samples represent maximum possible fluorescence). Error bars are given for mean ± SD, n = 4 (**p<0.01, Student's t-test; p-values for each variant are 0.0022 (E3C), 0.0055 (I163C), 0.0091 (E3C/S88C). (**C**) Distance between Arg3 and Leu164 (measured from the centroid of each residue), over 400 ns of MD simulations of PglC in a POPE membrane. (**D**) Activity of SUMO-PglC variants following crosslinking with bBBr, normalized to activity following treatment with vehicle. Error bars are given for mean ± SD, n = 3.
DOI: https://doi.org/10.7554/eLife.40889.015

homologs is a powerful tool for identifying interacting pairs of residues within a structure (*Balakrishnan et al., 2011*; *Ovchinnikov et al., 2014*), and in some cases has even be used to model unknown protein folds (*Ovchinnikov et al., 2017*). A reentrant topology is significantly more likely than a membrane-spanning one to facilitate interactions between the hydrophobic and soluble domains of a fold: indeed, covariance analyses of PglC previously identified several residues in the RMH that contact the globular domain (*Lukose et al., 2015*; *Ray et al., 2018*).

Based on the above criteria, several 'bitopic' proteins from *E. coli* were identified that show evidence of reentrant N-terminal hydrophobic domains (*Table 1* and *Supplementary file 4*). Three candidates – LpxM, LpxL and LpxP – are of particular interest. These enzymes catalyze the transfer of myristoyl, lauroyl, or palmitoyl groups, respectively, to Kdo$_2$-lipid IV to produce lipid A, the lipidic component of outer membrane lipopolysaccharide found in most Gram-negative bacteria; as such, they carry significant therapeutic relevance as antibiotic targets. LpxM, LpxL and LpxP belong to a large family of lipid A biosynthesis acyltransferases: over 4000 homologs of each sequence were identified during covariance analyses. Additionally, the three enzymes belong to an extensive and diverse superfamily of lysophospholipid acyltransferases, which comprises members from all three domains of life and also includes several families of enzymes involved in triglyceride biosynthesis (*Shi and Cheng, 2009*; *Takeuchi and Reue, 2009*). The biochemical function of these enzymes in lipid A biosynthesis dictates that the soluble C-terminal domain of each must localize to the

**Table 1.** Candidate reentrant domains identified among predicted bitopic proteins from *E. coli*. Sequences were selected from the Membranome database of 196 known and predicted bitopic folds from *E. coli*, based on the criteria described in the text. Sequences highlighted in red represent the predicted transmembrane domain (TMHMM Server v2.0, http://www.cbs.dtu.dk/services/TMHMM/).

| Name | Function | Sequence of N-terminus (residues 1–50) |
|------|----------|------------------------------------------|
| HofN | Putative DNA utilization protein | MNPPINFLPWRQQRRTAFL RFWLLMFVAPLLLAVGITLILRLTGSAEARI |
| LpxL | Lipid A biosynthesis lauroyltransferase | MTNLPKFSTALLHPRYWLTWLG IGVLWLVVQLPYPVIYRLGCGLGKLALR |
| LpxM | Lipid A biosynthesis myristoyltransferase | METKKNNSEYIPEFDKSFRHP RYWGAWLGVAAMAGIALTPPKFRDPILAR |
| LpxP | Lipid A biosynthesis palmitoleoyltransferase | MFPQCKFSREFLHPRYWLTWFGL GVLWLWVQLPYPVLCFLGTRIGAMARP |
| YihG | Probably acyltransferase | MANLLNKFIMTRILAAITLLLSIVLTIL VTIFCSVPIIIAGIVKLLLPVP |

DOI: https://doi.org/10.7554/eLife.40889.016

cytoplasmic face of the inner membrane; thus, the predicted transmembrane topology places the N-terminus in the periplasm. However, we noted that multiple positively-charged residues precede the hydrophobic domain in each sequence, making localization of the N-terminus to the periplasm less likely. In addition, the three hydrophobic domains contain polar (Gln, Thr) and multiple aromatic (Trp, Tyr) residues not typically found in the middle of transmembrane helices (*von Heijne, 2006*), as well as helix-breaking Pro and Gly residues. Finally, covariance analyses of LpxM, LpxL and LpxP identified several instances of covariance between residues in the hydrophobic and globular domains (*Supplementary file 4*), suggesting interactions between the two. On the basis of these observations, we hypothesized that LpxM, LpxL and LpxP adopt reentrant membrane topologies, rather than the predicted membrane-spanning ones.

Accordingly, we performed a SCAM analysis to determine the topology of the N-terminal domain in members of the lipid A biosynthesis acyltransferase family. LpxM was chosen of the three acyltransferases as it has the fewest native cysteines, making it most amenable to the SCAM method. LpxM has two native cysteines in the globular domain (C73 and C240), neither of which is highly conserved; thus, either one or both were substituted with serine to create unique cysteine variants C73 and C240 and a 'Cysless' variant. Unique cysteines were introduced to the 'Cysless' LpxM at non-conserved positions at the N-terminus (S8, I11 and S17) and at an additional location on the globular domain (M89). SCAM analysis of 'Cysless' LpxM and the six unique cysteine variants revealed that both the N-terminus and the globular domain of LpxM are located in the cytoplasm (*Figure 7A*). This confirms a reentrant topology for LpxM. Given the similarity between LpxM, LpxL and LpxP, we propose that the reentrant topology is conserved among the related lipid A biosynthesis acyltransferases. Notably, the only lipid A biosynthesis acyltransferase for which a structure has been reported to date is the LpxM homolog from *Acinetobacter baumannii* (*Dovala et al., 2016*). The N-terminus of the LpxM homolog from *A. baumannii* is similarly reported to be a predicted transmembrane domain; however, in the reported structure, this domain protrudes from the globular domain as a helix-break-helix (*Figure 7B*), reminiscent of a reentrant domain. We also noted that the hydrophobic region is shorter in this homolog than in those from *E. coli* and other Gram-negative pathogens (*Figure 7C*), making a membrane-spanning topology particularly unlikely. On the contrary, this could indicate that the N-terminus of LpxM from *A. baumannii* forms a reentrant domain that penetrates the membrane more shallowly than the reentrant domains of LpxM homologs from *E. coli* and other bacteria.

## Discussion

The recently reported structure of the monotopic PGT, PglC, describes a mode of membrane association reliant on a single reentrant helix-break-helix domain inserted in the membrane (*Ray et al., 2018*). Formation of this RMH is largely driven by two highly conserved motifs, elucidated in this

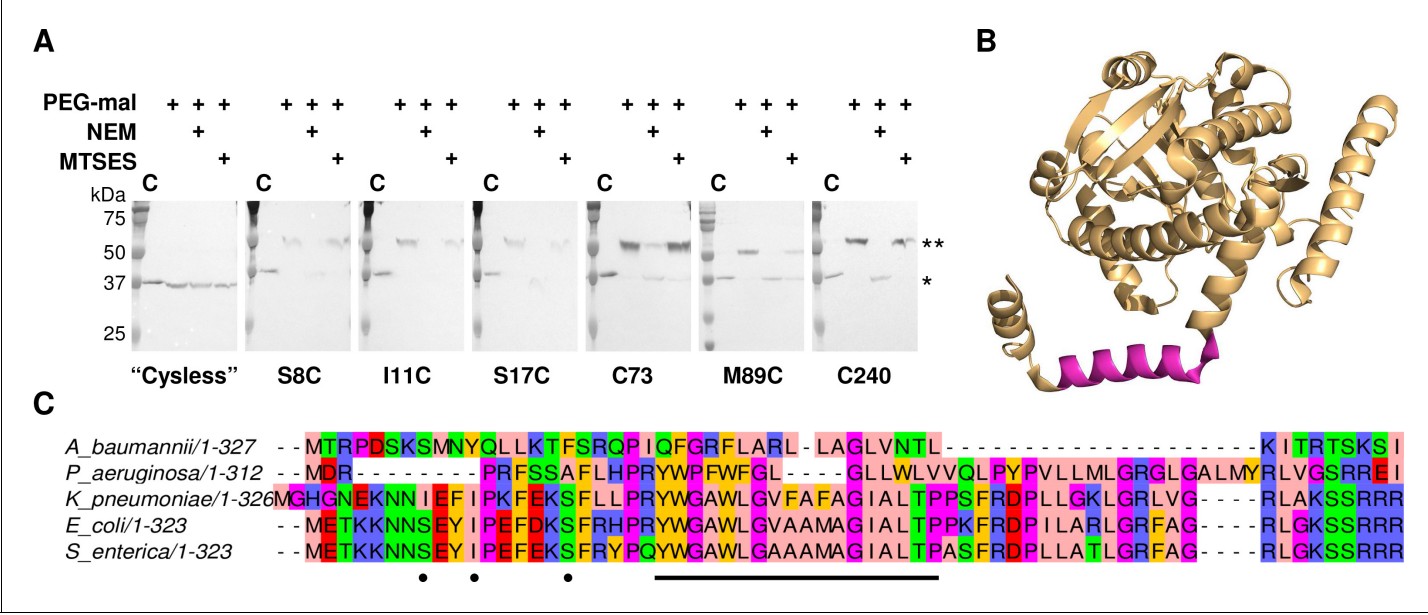

**Figure 7.** LpxM adopts a reentrant membrane topology. (**A**) SCAM analysis on LpxM from *E. coli* indicates that the fold adopts a reentrant membrane topology rather than the predicted membrane-spanning one (* = native LpxM; ** = LpxM labeled with PEG-mal; C = control, no PEG-mal labeling). All SCAM experiments were performed in duplicate or more. Representative Western blots are shown. (**B**) The structure of LpxM from *A. baumannii* (*Dovala et al., 2016*); PDB 5KN7. The predicted transmembrane domain, as reported for the structure, is shown in pink. (**C**) Sequence alignment of LpxM from *A. baumannii, Pseudomonas aeruginosa, Klebsiella pneumoniae, E. coli* and *Salmonella enterica*. Only the N-terminus is shown. The predicted transmembrane region, corresponding to residues 23–40 of LpxM from *E. coli*, is underlined with a black bar. Black dots indicate the location of unique cysteines introduced into the N-terminus of LpxM for SCAM analysis.

DOI: https://doi.org/10.7554/eLife.40889.017

work, that contribute to both early folding events that facilitate membrane insertion of the RMH in the proper topology, and late folding events that stabilize the folded PglC in the membrane. Insight from the in-depth study of PglC further enabled us to demonstrate that that the co-occurrence of similar motifs could be employed to identify reentrant topologies in other unrelated membrane proteins.

## Two conserved motifs contribute to RMH formation

The in vivo SCAM and in silico peptide folding experiments presented herein suggest a synergistic effect between the Lys7-Arg8 and Ser23-Pro24 motifs in determining proper RMH formation. The synergy observed between Ser23 and Pro24 echoes the intramolecular interactions observed both in the crystal structure and MD simulations of PglC, wherein Pro24 disrupts the backbone hydrogen-bonding network of the RMH helix to create the characteristic break, and the backbone amide and side chain of Ser23 stabilize this break by forming hydrogen bonds with the backbone carbonyls of Leu19 and Ile20.

Prolines are known to break α-helices by perturbing the peptide backbone hydrogen-bonding network that dictates α-helicity (*Piela et al., 1987*). Hydrogen bonding of the peptide backbone is thought to drive folding and membrane insertion of hydrophobic peptides by greatly reducing the energetic cost of partitioning peptide bonds into the membrane (*Cymer et al., 2015*; *Mackenzie, 2006*), wherein unsatisfied backbone hydrogen bonds are estimated to carry a free energy cost of 0.4 kcal mol$^{-1}$ per residue (*Almeida et al., 2012*). Consequently, the interruption in backbone hydrogen bonding introduced by Pro24 has a destabilizing effect on the RMH, which is largely mitigated by auxiliary hydrogen bonding between Ser23 and the peptide backbone carbonyl groups of Ile19 and Ile20. Polar residues such as serine in hydrophobic peptides were also identified previously as characteristic features of reentrant helices (*Yan and Luo, 2010*). Thus, the two residues of the Ser-Pro motif act cooperatively to define a modified hydrogen-bonding network in the RMH that yields the observed helix-break-helix structure. The absence of these key residues in S23A/P24A

PglC accordingly biases the N-terminus towards formation of a continuous α-helical secondary structure (*Figure 3B*), and could explain the aberrant topology observed for S23A/P24A PglC by SCAM analysis (*Figure 2C*).

Notably, a conserved Ile-Pro motif was previously identified as a key determinant of the reentrant loop topology of caveolin-1 (*Aoki et al., 2010*; *Lee and Glover, 2012*), and while Ser23 is 49% conserved among PglC homologs, Pro24 is otherwise often preceded by a branched-chain amino acid (BCAA): Ile, Leu, or Val. This suggests that BCAA-proline motifs may also be capable of supporting helix-break-helix formation among monotopic membrane proteins with reentrant helix domains.

Lys7 and Arg8 also act jointly with each other and with the Ser-Pro motif to enforce a reentrant topology, with the K7A/R8A and the K7A/R8A/P24A PglC variants adopting increasingly perturbed topologies. The observation that substitution of both conserved positive charges caused the N-terminus to lose some of its proclivity for cytoplasmic localization is in good agreement with previous reports of topology determination by the positive-inside rule (*Gafvelin and von Heijne, 1994*; *Nørholm et al., 2011*), but the observation that a significant portion of the K7A/R8A PglC population continued to adopt the native reentrant topology suggests that while Lys7 and Arg8 contribute to proper topology formation, they are not the primary drivers. More likely, the native reentrant PglC topology results from the cumulative effect of Lys7, Arg8, Ser23 and Pro24: each of these four conserved residues exerts a unique influence on formation and maintenance of the RMH domain to collectively result in the observed topology.

## RMH formation as an early folding event

Due to the hydrophobicity of the N-terminus of PglC, it may act as an uncleaved signal sequence (*Martoglio and Dobberstein, 1998*) to target the nascent polypeptide to the membrane early during PglC translation. Thus, formation and membrane insertion of the RMH domain could occur while the globular domain of PglC is still being synthesized (*Figure 8*). We propose that the two conserved motifs identified herein each play critical roles in this early folding event. The Lys7 and Arg8 at the N-terminus each contribute positive charges that disfavor localization of the N-terminus to the periplasm (*Martoglio and Dobberstein, 1998*), and, in agreement with the positive-inside rule, retain these residues at the cytoplasmic face of the inner membrane. As translation of the RMH continues, the modified backbone hydrogen-bonding pattern of the Ser-Pro motif facilitates formation of the characteristic helix-break-helix of the RMH and directs translation and folding of the globular domain to proceed on the cytoplasmic side of the membrane. Thus, under the combined influence of the Lys-Arg and Ser-Pro motifs, the hydrophobic domain of PglC inserts into the membrane as a reentrant helix, establishing a monotopic topology for the final fold. Accordingly, alanine-substitution of these key motifs likely leads to aberrant RMH formation and resulted in the non-native topologies observed by SCAM analysis (*Figure 2C* and *Figure 5*). Indeed, alanine-substitution results in such a disruption in the native PglC membrane insertion process that the entire construct appears to be erroneously translocated into the periplasm in a non-native conformation. A similar translocation of mistranslated proteins into the periplasm has previously been reported as a possible mechanism of action for aminoglycosidase toxicity in *E. coli* (*Kohanski et al., 2008*).

## PglC structure and function are supported by the two conserved motifs

Several lines of evidence suggest that both the Lys-Arg and Ser-Pro motifs additionally make significant contributions to maintaining the stability of the RMH within the context of fully-folded PglC to support function. Dibromobimane crosslinking of the N-terminus to the globular domain in the E3C/I163C SUMO-PglC variant indicates that the N-terminus can be captured in close proximity to the globular domain and that it can remain in such a conformation during catalysis (*Figure 6*). MD simulations additionally demonstrate that the N-terminus has little conformational freedom, supporting the hypothesis that crosslinking captures a stable conformation of the RMH. Taken together, these results strongly suggest that the reported structure represents a native and active conformation of the RMH in the PglC fold. In the reported structure of PglC, Lys7 forms a salt bridge with Asp169, a residue which is 98% conserved among PglC homologs, at the interface between the N-terminus and the globular domain (*Figure 1D*). The corresponding Asp in the PglC homolog from *C. jejuni* was reported previously to be necessary for catalytic activity (*Lukose et al., 2015*). The K7A SUMO-PglC variant was similarly found to be catalytically inactive, despite adopting a reentrant topology,

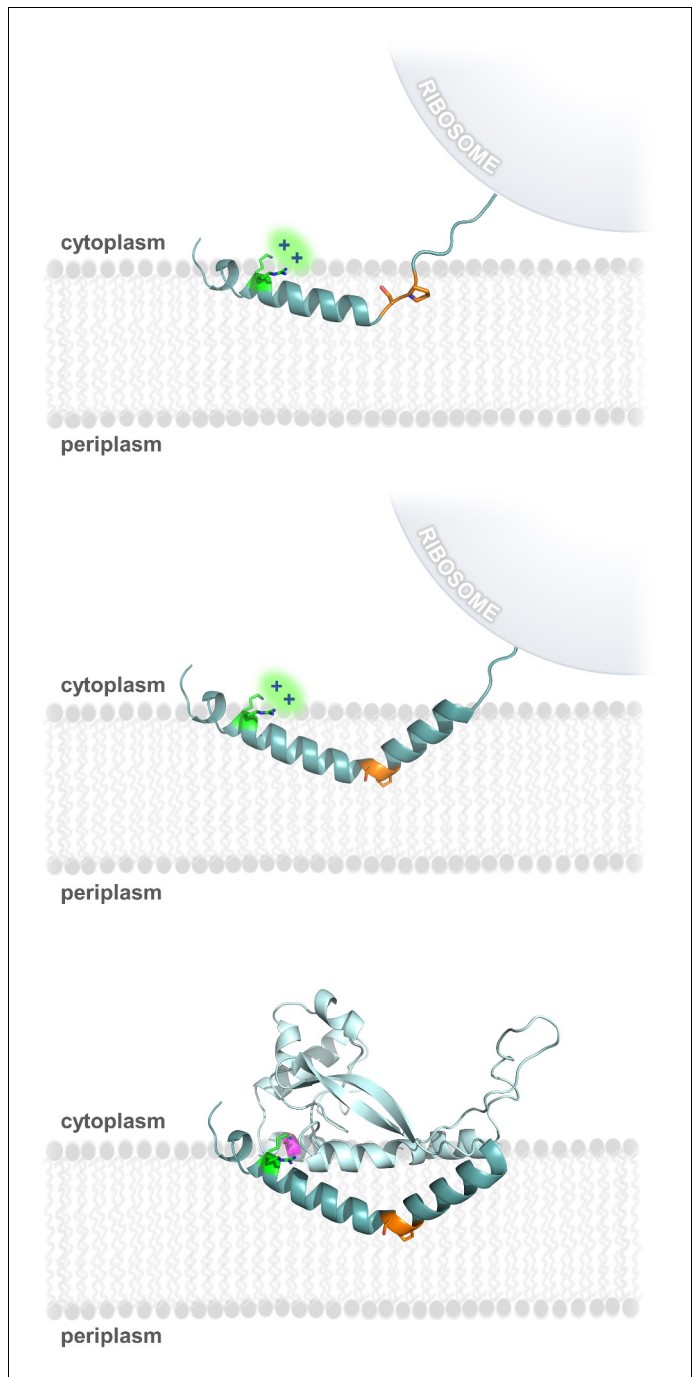

**Figure 8.** Model of RMH folding and membrane insertion. The Lys-Arg (green) and the Ser-Pro (orange) motifs facilitate formation of the RMH and insertion into the membrane in an early co-translational event. The positively-charged Lys-Arg motif favors localization of the N-terminus to the cytoplasm (top panel). The Ser-Pro motif creates the characteristic break in the RMH (middle panel), resulting in insertion of the RMH into the membrane in a reentrant topology. Following translation of the globular domain, both motifs further contribute to the overall stability of the PglC fold (bottom panel).

DOI: https://doi.org/10.7554/eLife.40889.018

as were K7A/R8A and K7A/R8A/P24A SUMO-PglC (*Figure 9*). This suggests that the salt bridge formed between Lys7 and Asp169 helps to constrain the N-terminus in close proximity to the globular domain, and that this interaction is necessary for catalytic activity. Thus, in addition to informing early RMH folding, Lys7 plays a crucial structural role by stabilizing the RMH in the PglC fold via a key interaction with Asp169 on the globular domain and thereby promoting PglC activity.

The contribution made by Arg8 to stability of PglC is more subtle. In the reported structure of PglC, a phosphatidylethanolamine head-group is coordinated to the guanidinium side chain of Arg8 (*Ray et al., 2018*). Previously, MD simulations of guanidinium ion translocation into the membrane reported a free energy minimum for the ion at the head-group region of the membrane-water interface (*Schow et al., 2011*). This suggests that interactions between Arg8 and surrounding lipid head-groups help stabilize the RMH at the membrane interface. Unlike the inactive K7A SUMO-PglC variant, R8A SUMO-PglC was found to retain ~15% catalytic activity (*Figure 9*). Thus, Arg8 may play a less significant role in the PglC fold than does Lys7.

The stability of PglC is also influenced by the Ser-Pro motif. We propose that the network of hydrogen bonds enforced by the motif (*Figure 1C* and *Figure 3A*) acts as a rigidifying 'staple' at the break in the RMH to restrict the conformational freedom of this domain. The rigidity conferred by the Ser-Pro motif would then allow proper positioning of the RMH domain with respect to the globular domain and formation of key intramolecular contacts (such as the salt bridge between Lys7 and Asp169), resulting in stabilization of the native PglC fold. The contribution made by the Ser-Pro motif is evidenced by the increased thermal stability of the wild-type SUMO-PglC relative to the S23A/P24A SUMO-PglC variant, and by MD simulations in which S23A/P24A PglC experienced a significant collapse into the fold interior relative to wild-type PglC (*Figure 4*). Thus, the Lys-Arg and Ser-Pro motifs, in addition to influencing formation of the RMH domain, play important roles in positioning the RMH within PglC to stabilize the overall fold and support PglC function.

As the only membrane-inserted domain in PglC, the RMH is likely responsible for binding to the polyprenol phosphate substrate in the membrane; Pro24 in particular is hypothesized to play an important role (*Lukose et al., 2015*). Conserved prolines are found in polyisoprene recognition sequences in various enzymes (*Zhou and Troy, 2003*), and were previously proposed to contribute to polyprenol binding, possibly as a result of their disruption of peptide backbone hydrogen bonding (*Zhou and Troy, 2005*). Indeed, a P24A SUMO-PglC variant was previously reported to be catalytically inactive (*Lukose et al., 2015*). Notably, although both S23A and P24A SUMO-PglC variants retain native-like reentrant topologies, the S23A variant exhibits near-wild type activity, whereas the P24A and S23A/P24A variants are inactive (*Figure 9*). This suggests that in addition to contributing to RMH formation, Pro24 in particular plays a crucial role in mediating polyprenol binding. Thus, in contributing to formation and stabilization of the RMH, the Lys-Arg and Ser-Pro motifs may enable function by positioning Pro24 to recognize polyprenol phosphate and encourage binding in a catalytically-relevant configuration relative to the PglC active site.

## Proposed guidelines for identifying reentrant domains in monotopic membrane proteins

The RMH domain is central to the structure and function of PglC; it anchors PglC in the membrane, interacts with several amphipathic helices to position the active site of PglC at the membrane-water interface, and mediates binding of the polyprenol phosphate substrate. The essentiality of the RMH to PglC structure and function, and the lack of homology to any known soluble protein folds, indeed suggest evolution of the fold at the membrane interface precisely to catalyze such transfer reactions. In this study, we identify four conserved residues that each make specific contributions both to the early formation of the RMH and to the stability of the final PglC fold, creating a clear preference for a reentrant topology for PglC and facilitating PglC function.

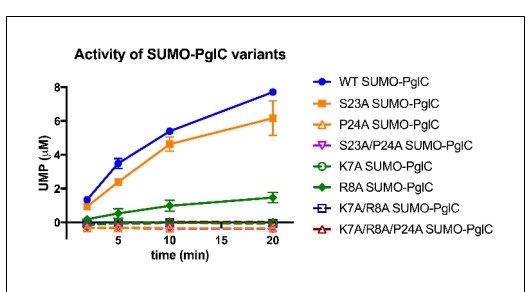

**Figure 9.** Individual residues differ in their importance for PglC function. Activity assays of wild-type SUMO-PglC and variants using UMP-Glo to monitor UMP release. Error bars are given for mean ± SD, n = 2. DOI: https://doi.org/10.7554/eLife.40889.019

The proposed model for RMH formation in PglC (*Figure 8*) builds on existing principles of membrane protein topogenesis; both the conserved positive charges of the Lys-Arg motif and the helix-breaking 'staple' of the Ser-Pro motif agree well with published observation of the positive-inside rule (*Gafvelin and von Heijne, 1994*; *Heijne, 1986*; *von Heijne, 2006*) and peptide backbone hydrogen bonding (*Cymer et al., 2015*; *Mackenzie, 2006*) as driving forces for folding and topology determination of membrane proteins. These key principles are also well-illustrated in the context of the PglC structure. A previous study of the topologies of caveolin and diacylglycerol kinase ε suggested a fine balance between reentrant and membrane-spanning topologies for both proteins, and noted that both positive flanking charges and conserved proline residues in these proteins could play decisive roles in topology determination (*Nørholm et al., 2011*); the formation of the PglC RMH similarly depends on both. However, unlike PglC, caveolin and diacylglycerol kinase ε have not been structurally characterized to date. The current study is significant as it is the first to frame these effects in the context of the experimentally-determined structure.

The two motifs highlighted by this study cooperatively enforce a reentrant helix-break-helix topology in PglC. Whereas few reentrant monotopic topologies have been structurally characterized to date, topology predications have suggested that bitopic membrane proteins constitute 15–25% of all integral membrane proteins bacteria and are even more prevalent among eukaryotes, accounting for almost half of all human membrane proteins (*Almén et al., 2009*; *Krogh et al., 2001*). As PglC was also previously predicted to adopt a bitopic topology, we investigated whether knowledge of topology determination in PglC might facilitate the identification of additional examples of membrane proteins that are predicted to be bitopic, but in fact adopt monotopic topologies similar to PglC.

Towards this goal, we exploited the Membranome database of single-helix transmembrane proteins (http://membranome.org/) (*Lomize et al., 2017*), which contains curated data on known and predicted bitopic membrane proteins from various genomes. The studies showed that LpxM, which is representative of long-chain fatty acid acyltransferases that catalyze acylation of $Kdo_2$-lipid IV to produce lipid A, could be reassigned as a monotopic membrane protein based on sequence motifs and complementary covariance and SCAM analyses. The example of LpxM underscores the importance of a deeper understanding of membrane topology determination, particularly with respect to single hydrophobic domains in monotopic and bitopic proteins. Whereas it is standard to annotate stretches of hydrophobic residues as membrane-spanning helices, the current study demonstrates that such helices are capable of adopting a greater diversity of topologies. The presence of positively-charged residues preceding, and helix-disrupting residues throughout, the hydrophobic domain appears to bias towards a reentrant topology. Such features can thus be taken as indicators of a reentrant topology, particularly if such a topology is compatible with biological function, as in the case for both PglC and LpxM. Covariance analysis can additionally be an invaluable tool for topology prediction, as it has the capacity to identify interactions between residue pairs that strongly support a reentrant topology. We note that, due to the large number of homologous sequences required for accurate covariance analysis, this tool is more amenable to bacterial proteins with homologs in many species for which genomic data are readily available. However, we anticipate that with the identification of additional examples of reentrant domains in membrane proteins from all domains of life, comprehensive prediction of reentrant topologies in eukaryotes as well as prokaryotes will become more accessible.

The insights gained in the presented study deepen our understanding of the many forces that dictate formation and stability of reentrant domains, particularly in monotopic membrane proteins. We expect that such domains are often misclassified as transmembrane domains and result in erroneous membrane topology predictions; we have shown this to be the case for both the monotopic PGT and lipid A biosynthesis acyltransferase extensive protein families. This work also provides generalizable guidelines for identifying reentrant topologies in membrane proteins of interest. We demonstrate that these guidelines can be applied in a manual parsing of 196 proteins from *E. coli* with predicted transmembrane domains, and successfully identify a family of enzymes that indeed adopts a reentrant, rather than the predicted membrane-spanning, topology. We anticipate that this work lays a foundation for computational methods to predict reentrant topologies more broadly in a diversity of membrane proteins.

## Materials and methods

### Molecular dynamics simulations of full-length PglC

All MD simulations were performed with Amber14 (*Case et al., 2014*) (RRID:SCR_014230) using the deposited PglC coordinates (PDB 5W7L).

#### Membrane insertion model

The placement of PglC in relation to the membrane was predicted using the PPM server: http://opm.phar.umich.edu/server.php (*Lomize et al., 2012*).

#### Simulations of PglC in a POPE membrane

Steepest descent gradient algorithm was iterated for 5000 steps followed by 5000 iterations of conjugate gradient algorithm under no constraint. The system was then heated from 0 to 100 K for 2500 steps in the NVT ensemble while the protein and the lipids were held by a 10 kcal mol$^{-1}$ Å$^{-2-}$ harmonic potential. In the subsequent step, the system was heated from 100 K to 303 K for 50,000 steps. In the membrane system, the dimensions of the box can change considerably during the first nanoseconds of simulation, thus to allow the program to recalculate them frequently the first 10 steps of the production run were performed for a maximum of 500 ps. In all steps the temperature was controlled by a Langevin thermostat. The heating phase and the production run were performed under an anisotropic NPT ensemble to account for different physical properties along the dimensions tangential to the membrane relative to the ones normal to the membrane.

#### Distance measurement over time

Distances were measured over time with cpptraj within AmberTools15 (*Case et al., 2015*). Hydrogen bonding was evaluated based on the distance threshold criterion of a H-O distance less than 2.5 Å.

#### In silico mutations

The in silico mutations of the Ser-Pro motif in the PglC structure were introduced with Modeller (*Sali and Blundell, 1993*) (RRID:SCR_008395) in two steps: Ser23 was mutated to Ala first, followed by mutation of Pro24 to Ala.

#### Comparison of wild-type and S23A/P24A PglC

Dynamics of wild-type and S23A/P24A PglC were compared in terms of backbone RMSD and RMS fluctuations per residues based on a minimum fit performed on the backbone of the protein. Snapshots of the structures along the simulations were also compared visually and rendered with PyMol 1.6.0.0 (*Schrodinger, 2015*) (RRID:SCR_000305).

### PglC variants

Wild-type PglC from *C. jejuni* strain 11168 was cloned into the pET24a vector using the NdeI and XhoI restriction sites to insert a C-terminal His$_6$-tag or into the pE-SUMO vector as reported previously (*Lukose et al., 2015*). Unique cysteines at the K4, F6, S88 and S186 positions (for SCAM) and at the E3 and I163 positions (for bBBr crosslinking) were introduced using QuikChange II Site-Directed Mutagenesis (Agilent Technologies, Santa Clara, CA) according to manufacturer's instructions. Alanine substitutions at K7, R8, S23 and P24 were all similarly introduced by Quikchange. All primers used for subcloning and mutagenesis are listed in *Supplementary file 2*.

### SCAM analysis

All SCAM analyses were performed as reported previously for wild-type PglC-His$_6$ (*Ray et al., 2018*). Biological replicates were performed starting with a common glycerol stock of each construct.

### Peptide folding

Two elongated peptide structures were generated using Leap, distributed within AmberTools15 (*Case et al., 2015*) (RRID:SCR_014230). Peptide sequences are provided in *Supplementary file 3*.

Leap was also used to generate solvent boxes and add counter ions. Peptides were placed into a water-solvated truncated octahedron simulation box with a minimum distance between the box border and the peptide of 5 Å. The aspartic acid was considered charged negatively, so the total charge was zeroed by adding a sodium ion. The two peptides were simulated for 100 ns each following the protocol described below. The last snapshot of each simulation was extracted and inserted into new truncated octahedron box solvated with a pre-equilibrated 20% isopropanol/water mixture retrieved from the literature (*Alvarez, 2012*). Both peptides were further simulated in this environment for an additional 1.5 μs.

## Peptide folding in water and in 20% isopropanol/water

The system underwent 1000 steps of steepest descent algorithm followed by 7000 steps of a conjugate gradient algorithm under a 100 kcal mol$^{-1}$Å$^{-2}$ harmonic potential constraint applied to the protein. The conjugate gradient algorithm minimization continued while the harmonic potential is progressively lowered to 10, 5, 2.5 and 0 kcal mol$^{-1}$Å$^{-1}$ every 600 steps. The system was then heated from 0 K to 100 K using the Langevin thermostat in the canonical ensemble (NVT) while a 20 kcal mol$^{-1}$Å$^{-2}$ harmonic potential restraint was applied on the protein. Finally, the system was heated from 100 K to 300 K in the isothermal–isobaric ensemble (NPT) under the same restraint conditions as the previous step, followed by a simulation for 100 ps under no harmonic restraint. At this point, the system was ready for the production run, which was performed using the Langevin thermostat under NPT ensemble, at a 2-fs time step.

## SUMO-PglC purification

SUMO-PglC variants were expressed in BL21-CodonPlus (DE3)-RIL *Escherichia coli* cells (Agilent Technologies) using the Studier auto-induction method (*Studier, 2005*). Overnight cultures grown in 3 mL MDG media (0.5% (w/v) glucose, 0.25% (w/v) sodium aspartate, 2 mM MgSO$_4$, 25 mM Na$_2$HPO$_4$, 25 mM KH$_2$PO$_4$, 50 mM NH$_4$Cl, 5 mM Na$_2$SO$_4$ and 0.2x trace metal mix (from 1000x stock, Teknova, Hollister, CA)) with kanamycin and chloramphenicol (30 μg/mL each) were used to inoculate 0.5 L auto-induction media (1% (w/v) tryptone, 0.5% (w/v) yeast extract, 0.5% (v/v) glycerol, 0.05% (w/v) glucose, 0.2% (w/v) α-D-lactose, 2 mM MgSO$_4$, 25 mM Na$_2$HPO$_4$, 25 mM KH$_2$PO$_4$, 50 mM NH$_4$Cl, and 5 mM Na$_2$SO$_4$, 0.2x trace metal mix) containing kanamycin (90 μg/mL) and chloramphenicol (30 μg/mL). Cells were grown for 4 hr at 37°C followed by an additional 16–18 hr at 16°C and then harvested at 3700 x *g* for 30 min.

All protein purifications were carried out at 4°C. Cells were re-suspended in 10% original culture volume in lysis buffer (50 mM HEPES, 150 mM NaCl, pH 7.5, supplemented with 0.5 mg/mL lysozyme (Research Products International, Mount Prospect, IL), 1:1000 dilution of EDTA-free protease inhibitor cocktail (EMD Millipore, Burlington, MA) and 1 unit/mL DNase I (New England Biolabs, Ipswich, MA)). Cells were lysed by sonication (Vibra-Cell, 50% amplitude, 1 s ON – 2 s OFF, 2 × 1.5 min; Sonics, Newtown, CT). Lysate was centrifuged at 9000 x *g* for 45 min at 4°C. The resulting supernatant was further centrifuged at 140,000 x *g* for 65 min at 4°C to yield the cell envelope fraction (CEF), which was homogenized into 2.5% original culture volume of solubilization buffer (50 mM HEPES, 100 mM NaCl, 1% DDM (*n*-dodecyl β-D-maltoside, Anatrace, Maumee, OH), pH 7.5, and additional protease inhibitor (1:1000 dilution)). The suspension was tumbled on a rotating mixer overnight at 4°C, after which it was centrifuged at 150,000 x *g* for 65 min to remove any insoluble material.

The supernatant was incubated with Ni-NTA resin (1 mL per L original culture volume; Thermo Fisher Scientific, Waltham, MA)) pre-treated with equilibration buffer (50 mM HEPES, 100 mM NaCl, 20 mM imidazole, 5% glycerol, pH 7.5) for 1 hr. The resin was washed with 20 column volumes of wash-1 buffer (equilibration buffer + 0.03% DDM), followed by 20 column volumes of wash-2 buffer (equilibration buffer + 0.03% DDM, 45 mM imidazole). The protein was eluted from the column in 2 column volumes of elution buffer (equilibration buffer + 0.03% DDM, 500 mM imidazole). Elution fractions were combined and immediately desalted using a 5 ml HiTrap desalting column (GE Healthcare, Marlborough, MA) that was pre-equilibrated with desalting buffer (50 mM HEPES, 100 mM NaCl, 0.03% DDM, 5% glycerol, pH 7.5). Fractions containing SUMO-PglC were supplemented with additional DDM to a final concentration of 0.2% and flash frozen for storage at −80°C.

## Circular dichroism

Circular dichroism was performed on a JASCO Model J-1500 Circular Dichroism Spectrometer. Spectral scans were performed with 16 µM purified SUMO-PglC in phosphate buffer (20 mM $Na_2HPO_4$, 100 mM NaCl, 5% glycerol, 0.2% DDM, pH 7.5). Readings were taken at 4°C from 190 to 250 nm at 0.5 nm intervals.

## Thermal shift assay

Thermal shift assays were based on a protocol described previously (*Gandini et al., 2017*). Aliquots of 6–8 µM purified SUMO-PglC in PglC buffer (50 mM HEPES, 100 mM NaCl, 5% glycerol, 0.2% DDM, pH 7.5) were heated for 10 min at 30–99°C in a PCR machine (MJ Mini Thermal Cycler; Bio-Rad, Hercules, CA). Precipitate was immediately removed by centrifugation at 16,000 x *g* for 10 min at 4°C. The resulting supernatant, containing protein that remained soluble, was analyzed by SDS-PAGE with Coomassie staining and quantified by gel densitometry using a Molecular Imager Gel Doc XR+ System with Image Lab software (BioRad). Data were fitted using Graphpad Prism 7 (GraphPad Software, La Jolla, CA; RRID:SCR_002798). Technical replicates were performed on distinct aliquots taken from a common protein purification prep.

## Dibromobimane crosslinking

For crosslinking experiments, 5 µM SUMO-PglC in PglC buffer (50 mM HEPES, 100 mM NaCl, 5% glycerol, 0.2% DDM, pH 7.5) was treated with 25 µM dibromobimane (Sigma-Aldrich, St. Louis, MO) from a 1 mM stock in 20% acetonitrile for 20 min in the dark at room temperature. For fluorescence analysis, some crosslinked samples were further treated with 50 mM DTT for 1 min, and all samples were then analyzed by SDS-PAGE and quantified by gel densitometry using a Molecular Imager Gel Doc XR+ System with Image Lab software (BioRad). Fluorescence was measured using the ethidium bromide excitation setting (302 nm) and normalized to band intensity after Coomassie staining. Relative fluorescence of SUMO-PglC variants is reported as fluorescence intensity, normalized to Coomassie staining, relative to DTT-quenched samples. Technical replicates were performed on distinct aliquots taken from a common protein purification prep.

For activity assays crosslinked SUMO-PglC prepared as above was diluted with PglC buffer to a 50 nM stock and assayed as described below. Activity of variants is reported relative to control samples that were treated with carrier (20% acetonitrile) and assayed in parallel with crosslinked samples. Data were plotted using Graphpad Prism 7 (GraphPad Software, RRID:SCR_002798).

## SUMO-PglC activity assay

SUMO-PglC activity assays were performed as described previously using the UMP/CMP-Glo assay (Promega, Madison, WI) (*Das et al., 2017*; *Das et al., 2016*). Assays were performed on a 10 µL scale at room temperature with 5 nM enzyme (from a 50 nM stock) and 20 µM of both UDP-N,N'-diacetylbacillosamine and Und-P substrates (from 200 µM stocks in water and DMSO, respectively) in assay buffer (50 mM HEPES, 100 mM NaCl, 5 mM $MgCl_2$, 0.1% Triton X-100, pH 7.5; in assays of Cys-containing SUMO-PglC variants, 7 mM β-mercaptoethanol was also added to the assay buffer). Following quenching with 10 µL of UMP-Glo reagent, luminescence was read in a 96-well plate (Corning, Inc., Corning, NY) using a SynergyH1 multimode plate reader (Biotek, Winooski, VT). The 96-well plate was shaken inside the plate reader chamber at 237 cpm at 25°C in the double orbital mode for 16 min, followed by 44 min incubation at the same temperature, after which time the luminescence was recorded (gain: 200, integration time: 0.5 s). Conversion of luminescence to UMP concentration was carried out using a standard curve. Data were plotted using Graphpad Prism 7 (GraphPad Software, RRID:SCR_002798). Technical replicates were performed on distinct aliquots taken from a common protein purification prep.

## Identification of candidate reentrant helical domains from the Membranome database

The Membranome database of single-helix transmembrane proteins (*Lomize et al., 2017*) was parsed manually for candidates with a reentrant topology. Candidates were selected from the available list of 196 putative bitopic proteins from *E. coli*; only proteins longer than 100 residues were considered. A preliminary list of candidates was composed of sequences in which the predicted

transmembrane domain was at the N-terminus (within the first 50 residues), was preceded by positively charged residues, and contained polar or charged, large aromatic (Trp/Tyr) and/or helix-breaking (Pro/Gly) residues. These sequences were then submitted to the GREMLIN webserver (http://gremlin.bakerlab.org/) (*Balakrishnan et al., 2011*) for multiple sequence alignment and covariance analysis. Notably, whereas a multiple sequence alignment was always returned, a covariance analysis was only performed by the server if enough homologs were identified to allow for accurate analysis. A final list of candidates, shown in *Table 1*, contained only those for which a) the N-terminal positively charged residues and helix-disrupting residues identified previously were found to be well represented among homologs in the multiple sequence alignment, b) a reentrant topology would be consistent with the biological role, and c) covariance analysis identifies interactions with >80% probability between residues at the N-terminus or within the hydrophobic domain and residues in the globular domain.

## LpxM variants

LpxM from *E. coli* K12 was synthesized and cloned into the pET24a vector by GenScript (Piscataway, NJ), using the NdeI and XhoI restriction sites to insert a C-terminal His$_6$-tag. Serine substitutions at C73 and C240, to give a 'Cysless' variant, were introduced using QuikChange II Site-Directed Mutagenesis (Agilent Technologies, Santa Clara, CA) according to manufacturer's instructions. Unique cysteines at S8, I11, S17 and M89, for SCAM analysis, were all similarly introduced by Quikchange. All primers used for mutagenesis are listed in *Supplementary file 2*.

# Acknowledgements

The Biophysical Instrumentation Facility for the Study of Complex Macromolecular Systems (NSF-0070319) is gratefully acknowledged for assistance with CD experiments. We thank Prof. Karen N. Allen for many valuable discussions, and Theresa Hwang and Hannah Bernstein for technical assistance with SCAM and activity analyses.

# Additional information

## Funding

| Funder | Grant reference number | Author |
|---|---|---|
| National Institutes of Health | NIH GM-039334 | Sonya Entova Barbara Imperiali |
| Ministerio de Economía y Competitividad | CTQ2014-57141-R | Jean-Marc Billod Sonsoles Martín-Santamaría |
| Jane Coffin Childs Memorial Fund for Medical Research | | Jean-Marie Swiecicki |
| National Institutes of Health | T32-GM007287 | Sonya Entova |
| Ministerio de Economía y Competitividad | CTQ2017-88353-R | Jean-Marc Billod Sonsoles Martín-Santamaría |

The funders had no role in study design, data collection and interpretation, or the decision to submit the work for publication.

## Author contributions

Sonya Entova, Conceptualization, Data curation, Formal analysis, Validation, Methodology, Writing—original draft, Writing—review and editing; Jean-Marc Billod, Data curation, Software, Formal analysis, Validation, Writing—original draft, Writing—review and editing; Jean-Marie Swiecicki, Conceptualization, Writing—review and editing; Sonsoles Martín-Santamaría, Conceptualization, Supervision, Writing—review and editing; Barbara Imperiali, Conceptualization, Supervision, Funding acquisition, Validation, Writing—original draft, Project administration, Writing—review and editing

Author ORCIDs

Sonya Entova (iD) http://orcid.org/0000-0002-5270-3336

Jean-Marc Billod (iD) https://orcid.org/0000-0002-3293-6378

Jean-Marie Swiecicki (iD) http://orcid.org/0000-0002-7139-8621

Sonsoles Martín-Santamaría (iD) http://orcid.org/0000-0002-7679-0155

Barbara Imperiali (iD) http://orcid.org/0000-0002-5749-7869

Decision letter and Author response

Decision letter https://doi.org/10.7554/eLife.40889.026

Author response https://doi.org/10.7554/eLife.40889.027

## Additional files

### Supplementary files

• Supplementary file 1. Corresponding key residues in *C. jejuni* and *C. concisus* PglC

DOI: https://doi.org/10.7554/eLife.40889.020

• Supplementary file 2. Primers used for cloning and mutagenesis of PglC and LpxM variants

DOI: https://doi.org/10.7554/eLife.40889.021

• Supplementary file 3. Amino acid sequences for peptide folding

DOI: https://doi.org/10.7554/eLife.40889.022

• Supplementary file 4. Results of covariance analyses performed on sequences in *Table 1* using the GREMLIN web-server (http://gremlin.bakerlab.org/). Interactions between residues of the hydrophobic and globular domains are highlighted in red (>90% probability) and orange (80–90% probability).

DOI: https://doi.org/10.7554/eLife.40889.023

• Transparent reporting form

DOI: https://doi.org/10.7554/eLife.40889.024

### Data availability

All data generated or analyzed during this study are included in the manuscript and supporting files.

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
