## [Decision Letter]

[Editors’ note: a previous version of this study was rejected after peer review, but the authors submitted for reconsideration. The first decision letter after peer review is shown below.]

Thank you for submitting your work entitled "Two conserved structural motifs determine membrane topology of a monotopic phosphoglycosyl transferase" for consideration by *eLife*. Your article has been reviewed by three peer reviewers, including Volker Dötsch as the Reviewing Editor and Reviewer #3, and the evaluation has been overseen by a Reviewing Editor and a Senior Editor. The following individual involved in review of your submission has agreed to reveal their identity: Richard Epand (Reviewer #2).

Our decision has been reached after consultation between the reviewers. Based on these discussions and the individual reviews below, we decided that the manuscript is not suitable for *eLife* in its current form, and we are returning the manuscript to you. However, all reviewers agree that the investigations of monotopic membrane proteins and the rules that define their topology are very important. They also agree that the study is technically of high quality and the paper is well written. The main criticism is that your study is too close to the structure paper already published. While it provides a nice analysis of the motifs that are important for a monotonic conformation, it is not clear if these motifs constitute a general rule or show an effect only in the context of PglC. Although you discuss with diacyl glycerol kinase epsilon and caveolin two potential other proteins with a similar topology, the manuscript would be greatly strengthened if you could provide experimental evidence with a peptide derived from these two proteins that the rules you identified for PglC are more generally applicable.

Reviewer #1:

In this contribution Entova et al., follow up a previous very recent contribution of the group (Ray et al., 2018, in press) to identify and characterize two conserved motifs that affect monotopic insertion of the protein PglC into the membrane and act cooperatively in the folding of the protein. They are related to membrane recognition and helix kink, so the protein can be properly accommodated in the cytoplasmic face of the membrane. A set of experiments and molecular dynamic simulations support the findings and the conclusions reported. The experiments seem to be adequately done and the paper is well written.

The system of study is utterly interesting, and the membrane insertion mechanism is unprecedented but most of the findings are already presented in the above-mentioned reference and the results included in this contribution add only incremental value to the project. Specifically, the high-resolution structure immediately reveals the significance of the Ser23-Pro24 motif while the other motif (Lys7-Arg8) is also mentioned in the Nat Chem Biol paper. Actually, most of Figure 1 and Figure 2B are based on previously published results.

More technical and aspects that should also be taken in consideration:

1) It is stated that S23A/P24A SUMO-PglC does not purify to sufficient homogeneity for reliable ellipticity measurements. Yet, the spectrum shown in Figure 4—figure supplement 2 is almost identical to the WT protein. The CD denaturation should be, at least, attempted and compared to the gel densitometry data, which is less accurate.

2) The destabilization (of 7.3 ºC) produced by the mutation is attributed to a particular destabilizing effect of the motif. Yet, destabilization upon mutation is a very frequent mechanism. Some controls (with alanine scanning mutations in other positions) are required to compare and contextualize this stability loss. In the same context, the aggregates should be structurally analyzed (i. e. electron microscopy) to determine to which extent the destabilization is intrinsic to the amino acid substitution (and not to the specific location).

Reviewer #2:

This manuscript describes the formation of a re-entrant helix to make this phosphoglycosyl transferase a monotopic protein. The authors use both experimental and computational methods to define two conserved motives in PglC, a phosphoglycosyl transferase from *Campylobacter jejuni*. A helix-breaking Ser-Pro motif facilitates the bending of a hydrophobic helix to allow it to leave the membrane on the same side that it entered and hence be a re-entrant helix. The results are put in context of other membrane protein helices. It is pointed out that re-entrant helices are under represented and likely play an important role in several membrane proteins. In the case of this protein it is also shown that the structure contributes to the stability of the protein and is important for its activity.

Reviewer #3:

Entova et al., describe a detailed characterization of two different motifs that they show are important for monotopic topologies of membrane proteins. The analysis is based on a previously published crystal structure of phosphoglycosyl transferase of *Campylobacter jejuni*. They show that two motifs, a stretch of positively charged amino acids at the N-terminus and a Ser-Pro in the middle of the embedded helix is important for the reentry topology observed in the structure. Through a combination of MD simulations and mutational analysis in combination with cysteine modification assays they show the importance of these regions.

The motivation of this study is to identify sequence motifs that allow the identification of reentry helices. The analysis of the authors clearly demonstrates the importance of these two motifs, although the results of some of the mutation studies cannot be interpreted. However, the analysis mainly deals with the detailed characterization of the previously published structure and does not extend significantly beyond the phosphoglycosyl transferase family. This is somewhat disappointing as it does not allow to define rules for other proteins. The Discussion section is also more a repetition of the Results section with little comparison to other proteins/studies.

The chemical cross linking studies might also not reflect the average topology of the helix as chemical cross linking is known to be able to capture only transient conformations (those ones with the correct distance). This should have been discussed.

---

## [Author Response]

[Editors’ note: the author responses to the first round of peer review follow.]

Our decision has been reached after consultation between the reviewers. Based on these discussions and the individual reviews below, we decided that the manuscript is not suitable for eLife in its current form, and we are returning the manuscript to you. However, all reviewers agree that the investigations of monotopic membrane proteins and the rules that define their topology are very important. They also agree that the study is technically of high quality and the paper is well written. The main criticism is that your study is too close to the structure paper already published. While it provides a nice analysis of the motifs that are important for a monotonic conformation, it is not clear if these motifs constitute a general rule or show an effect only in the context of PglC. Although you discuss with diacyl glycerol kinase epsilon and caveolin two potential other proteins with a similar topology, the manuscript would be greatly strengthened if you could provide experimental evidence with a peptide derived from these two proteins that the rules you identified for PglC are more generally applicable.

The major revision that we have made is to leverage the insights gained into monotopic topology determination in PglC for the identification of an important enzyme family that was previously predicted to be membrane-spanning (Discussion section). Specifically, the Membranome database of single-helix transmembrane proteins (http://membranome.org/) contains 196 bacterial sequences (all from *E. coli*) known or predicted to be bitopic. We parsed this list manually for sequences with N-terminal domains that, based on criteria defined in our studies on PglC, we predicted to actually be candidate monotopic proteins. Three of five identified sequences belong to a very extensive family of lipid A biosynthesis acyltransferases, functioning to transfer lauroyl, myristoyl and palmitoyl groups. We then applied the SCAM topology analysis method and confirmed that the N-termini of these folds indeed adopt reentrant topologies, contrary to previous predictions that these sequences were membrane-spanning. Thus, we anticipate that the drivers of topology determination for PglC, as identified in the manuscript, represent generally applicable rules that can, in fact, be applied to diverse monotopic folds.

Reviewer #1:[…] The system of study is utterly interesting, and the membrane insertion mechanism is unprecedented but most of the findings are already presented in the above-mentioned reference and the results included in this contribution add only incremental value to the project. Specifically, the high-resolution structure immediately reveals the significance of the Ser23-Pro24 motif while the other motif (Lys7-Arg8) is also mentioned in the Nat Chem Biol paper. Actually, most of Figure 1 and Figure 2B are based on previously published results.

We note that the X-ray structure, while intriguing, represents only a single static snapshot. It provides limited information on protein dynamics and on the key determinants of the unusual, yet potentially widespread, membrane-association mode that might be encrypted in the sequences of other currently undefined membrane protein structures. Additionally, the structure determination was carried out on a detergent-solubilized sample and thus in a non-native environment. Despite these limitations, insight from new structure provides both inspiration and guidance on important next steps to determine how features of the protein primary sequence lead to the previously uncharacterized monotopic structure during membrane topogenesis. In our revised version of the manuscript, we show that in addition to being widespread across the monotopic PGT superfamily, the defined primary sequence features can be used to identify similarly reentrant topologies in other monotopic folds (Discussion section). Thus, the content of the current manuscript is entirely new and absolutely will help to forge into new directions.

Concerning the figures, we note that Figure 1 accompanies the introduction and shows the sequence logo and the molecular detail of the motifs that are being examined in the paper (panels B, C, and D), so that the reader can visualize the goals of the work. These Pymol images are not claimed to be new information. Figures 1A and (previously) 1E are new and present the model of PglC in the membrane based on MD simulations as well as insight into the dynamical behavior of the H-bonding network at the Ser-Pro motif. Figure 1E has been recast as Figure 3A to distinguish it further as data new to the current study. Regarding Figure 2B, this single set of blots is provided specifically as context for the *new* data (that is precisely why it is separated out from the other data) to enable the reader to compare the behavior of all the new variants presented in this study, as shown in Figure 2C and Figure 5 (seven variants in total). The extensive studies, both computational and biochemical (Figure 3, Figure 4, Figure 5, Figure 6 and Figure 8 and newly added Figure 9, and the proposed model Figure 7) are all previously unpublished.

More technical and aspects that should also be taken in consideration:1) It is stated that S23A/P24A SUMO-PglC does not purify to sufficient homogeneity for reliable ellipticity measurements. Yet, the spectrum shown in Figure 4—figure supplement 2 is almost identical to the WT protein. The CD denaturation should be, at least, attempted and compared to the gel densitometry data, which is less accurate.

As demonstrated in Figure 4—figure supplement 2, whereas wild-type SUMO-PglC purifies to homogeneity, the S23A/P24A SUMO-PglC variant co-purifies with many contaminants. Therefore, although the CD spectra look similar, observing thermal melts using CD would not accurately reflect the melting behavior of S23A/P24A SUMO-PglC, when compared to the much-purer wild-type construct. We have amended the text to more clearly articulate these considerations (subsection “The Ser-Pro motif contributes to the overall stability of the PglC fold”) and have removed to spectra from Figure 4—figure supplement 2 to mitigate any confusion.

We have also included in the new text an additional motivation, not articulated in the previous version, for use of the thermal shift assay to measure stability: it has been noted in the literature that unfolding of α-helical membrane proteins is often driven through loss of tertiary contacts rather than secondary structure. Thus, CD denaturation studies, based on loss of secondary structure, are less likely to capture unfolding and thermal destabilization than the thermal shift assay.

2) The destabilization (of 7.3 ºC) produced by the mutation is attributed to a particular destabilizing effect of the motif. Yet, destabilization upon mutation is a very frequent mechanism. Some controls (with alanine scanning mutations in other positions) are required to compare and contextualize this stability loss. In the same context, the aggregates should be structurally analyzed (i. e. electron microscopy) to determine to which extent the destabilization is intrinsic to the amino acid substitution (and not to the specific location).

Controls with double alanine mutations in two additional positions have now been included in the manuscript as Figure 4—figure supplement 3 (subsection “The Ser-Pro motif contributes to the overall stability of the PglC fold”). We note that while these control mutations do result in a slight loss of thermal stability (2.5-3.5 *°*C), the effect is much less striking than the 7.3 *°*C destabilization due to mutation of the Ser-Pro motif. Thus, we maintain that the thermal shift assays of the Ser23A/P24A SUMO-PglC variant support a role for the Ser-Pro motif in maintaining stability of the PglC fold. With regard to the suggestion of alanine scanning mutagenesis, we note that covariance analysis was previously performed on 15,000 sequences from the monotopic PGT superfamily to identify contact pairs important to the PglC structure; we consider such an analysis very comprehensive and informative.

Concerning analysis of aggregates using EM we believe that the analysis would be complicated to interpret and that the study of the additional double alanine variants should be more reliable.

Reviewer #3:Entova et al., describe a detailed characterization of two different motifs that they show are important for monotopic topologies of membrane proteins. The analysis is based on a previously published crystal structure of phosphoglycosyl transferase of Campylobacter jejuni. They show that two motifs, a stretch of positively charged amino acids at the N-terminus and a Ser-Pro in the middle of the embedded helix is important for the reentry topology observed in the structure. Through a combination of MD simulations and mutational analysis in combination with cysteine modification assays they show the importance of these regions.The motivation of this study is to identify sequence motifs that allow the identification of reentry helices. The analysis of the authors clearly demonstrates the importance of these two motifs, although the results of some of the mutation studies cannot be interpreted. However, the analysis mainly deals with the detailed characterization of the previously published structure and does not extend significantly beyond the phosphoglycosyl transferase family. This is somewhat disappointing as it does not allow to define rules for other proteins. The Discussion section is also more a repetition of the Results section with little comparison to other proteins/studies.

As presented in the introduction, we have now extended the study to include the definition of guidelines for predicting comparable, reentrant topologies in other membrane proteins. In the amended manuscript, we show that we can leverage the insights gained from the in-depth analysis of PglC in the context of the X-ray structure to identify an entirely unrelated family of enzymes that were previously believed to be bitopic, but can be reclassified to have reentrant N-terminal domains.

The chemical cross linking studies might also not reflect the average topology of the helix as chemical cross linking is known to be able to capture only transient conformations (those ones with the correct distance). This should have been discussed.

As discussed in response to reviewer #1, while it is certainly possible for crosslinking to capture transient conformations, we believe that the additional computational studies and biochemical experiments described provide strong evidence to support that the crosslinked state is a stable, native and active conformation of PglC.